# Towards a Better Theoretical Understanding of Independent Subnetwork Training

## Abstract

Modern advancements in large-scale machine learning would be impossible without the paradigm of data-parallel distributed computing. Since distributed computing with large-scale models imparts excessive pressure on communication channels, a lot of recent research was directed towards co-designing communication compression strategies and training algorithms with the goal of reducing communication costs. While pure data parallelism allows better data scaling, it suffers from poor model scaling properties. Indeed, compute nodes are severely limited by memory constraints, preventing further increases in model size. For this reason, the latest achievements in training giant neural network models rely on some form of model parallelism as well. In this work, we take a closer theoretical look at Independent Subnetwork Training (IST), which is a recently proposed and highly effective technique for solving the aforementioned problems. We identify fundamental differences between IST and alternative approaches, such as distributed methods with compressed communication, and provide a precise analysis of its optimization performance on a quadratic model.

## 1 Introduction

A huge part of today's machine learning success drives from the possibility to build more and more complex models and train them on increasingly larger datasets. This fast progress has become feasible due to advancements in distributed optimization, which is necessary for proper scaling when the training data sizes grow [50]. In a typical scenario data parallelism is used for efficiency which consists of sharding the dataset across computing devices. This allowed very efficient scaling and accelerating of training moderately sized models by using additional hardware [19]. Though, such data parallel approach can suffer from communication bottleneck, which sparked a lot of research on distributed optimization with compressed communication of the parameters between nodes [3, 27, 38].

### 1.1 The need for model parallel

Despite the efficiency gains of data parallelism, it has some fundamental limitations when it comes to scaling up the model size. As the model dimension grows, the amount of memory required to store and update the parameters also increases, which becomes problematic due to resource constraints on individual devices. This has led to the development of model parallelism [11, 37], which splits a large model across multiple nodes, with each node responsible for computations of model parts [15, 47]. However, naive model parallelism also poses challenges because each node can only update its portion of the model based on the data it has access to. This creates a need for a very careful management of communication between devices. Thus, a combination of both data and model parallelism is often necessary to achieve efficient and scalable training of huge models.

Submitted to 37th Conference on Neural Information Processing Systems (NeurIPS 2023). Do not distribute.

---

**Algorithm 1** Distributed Submodel (Stochastic) Gradient Descent

---

1: **Parameters:** learning rate $\gamma > 0$; sketches $\mathbf{C}_1, \ldots, \mathbf{C}_n$; initial model $x^0 \in \mathbb{R}^d$
2: **for** $k = 0, 1, 2 \ldots$ **do**
3:   Select submodels $w_i^k = \mathbf{C}_i^k x^k$ for $i \in [n]$ and broadcast to all computing nodes
4:   **for** $i = 1, \ldots, n$ in parallel **do**
5:     Compute local (stochastic) gradient w.r.t. submodel: $\mathbf{C}_i^k \nabla f_i(w_i^k)$
6:     Take (maybe multiple) gradient descent step $z_i^+ = w_i^k - \gamma \mathbf{C}_i^k \nabla f_i(w_i^k)$
7:     Send $z_i^+$ to the server
8:   **end for**
9:   Aggregate/merge received submodels: $x^{k+1} = \frac{1}{n} \sum_{i=1}^{n} z_i^+$
10: **end for**

---

**IST.** Independent Subnetwork Training (IST) is a technique which suggests dividing the neural network into smaller independent sub-parts, training them in a distributed parallel fashion and then aggregating the results to update the weights of the whole model. According to IST, every subnetwork is operational on its own, has fewer parameters than the full model, and this not only reduces the load on computing nodes but also results in faster synchronization. A generalized analog of the described method is formalized as an iterative procedure in Algorithm 1. This paradigm was pioneered by [45] for networks with fully-connected layers and was later extended to ResNets [14] and Graph architectures [43]. Previous experimental studies have shown that IST is a very promising approach for various applications as it allows to effectively combine data with model parallelism and train larger models with limited compute. In addition, [28] performed theoretical analysis of IST for overparameterized single hidden layer neural networks with ReLU activations. The idea of IST was also recently extended to the federated setting via an asynchronous distributed dropout [13] technique.

**Federated Learning.** Another important setting when the data is distributed (due to privacy reasons) is Federated Learning [22, 27, 31]. In this scenario computing devices are often heterogeneous and more resource-constrained [5] (e.g. mobile phones) in comparison to data-center setting. Such challenges prompted extensive research efforts into selecting smaller and more efficient submodels for local on-device training [2, 6, 8, 12, 20, 21, 29, 35, 42, 44]. Many of these works propose approaches to adapt submodels, often tailored to specific neural network architectures, based on the capabilities of individual clients for various machine learning tasks. However, there is a lack of comprehension regarding the theoretical properties of these methods.

## 1.2 Summary of contributions

When reviewing the literature, we have found that a rigorous understanding of IST convergence virtually does not exist, which motivates our work. The main contributions of this paper include

- A novel approach to analyzing distributed methods that combine data and model parallelism by operating with sparse submodels for a quadratic model.
- The first analysis of independent subnetwork training in homogeneous and heterogeneous scenarios without restrictive assumptions on gradient estimators.
- Identification of settings when IST can optimize very efficiently or converge not to the optimal solution but only to an irreducible neighborhood which is also tightly characterized.
- Experimental validation of the proposed theory through carefully designed illustrative experiments. Due to space limitations, the results (and proofs) are provided in the Appendix.

## 2 Formalism and Setup

We consider the standard optimization formulation of distributed/federated learning problem [41],

$$\min_{x \in \mathbb{R}^d} \left[ f(x) := \frac{1}{n} \sum_{i=1}^{n} f_i(x) \right], \tag{1}$$

where $n$ is the number of clients/workers, each $f_i : \mathbb{R}^d \to \mathbb{R}^d$ represents the loss of the model parameterized by vector $x \in \mathbb{R}^d$ on the data of client $i$.

A typical Stochastic Gradient Descent (SGD) type method for solving this problem has the form

$$x^{k+1} = x^k - \gamma g^k, \qquad g^k = \frac{1}{n} \sum_{i=1}^{n} g_i^k, \qquad (2)$$

where $\gamma > 0$ is a stepsize and $g_i^k$ is a suitably constructed estimator of $\nabla f_i(x^k)$. In the distributed setting, computation of gradient estimators $g_i^k$ is typically performed by clients, sent to the server, which subsequently performs aggregation via averaging $g^k = \frac{1}{n} \sum_{i=1}^{n} g_i^k$. The result is then used to update the model $x^{k+1}$ via a gradient-type method (2), and at the next iteration the model is broadcast back to the clients. The process is repeated iteratively until a model of suitable qualities is found.

One of the main techniques used to accelerate distributed training is lossy *communication compression* [3, 27, 38]. It suggests applying a (possibly randomized) lossy compression mapping $\mathcal{C}$ to a vector/matrix/tensor $x$ before it is transmitted. This saves bits sent per every communication round at the cost of transmitting a less accurate estimate $\mathcal{C}(x)$ of $x$. The error caused by this routine also causes convergence issues, and to the best of our knowledge, convergence of IST-based techniques is for this reason not yet understood.

**Definition 1** (Unbiased compressor). *A randomized mapping* $\mathcal{C} : \mathbb{R}^d \to \mathbb{R}^d$ *is an* **unbiased compression operator** *(* $\mathcal{C} \in \mathbb{U}(\omega)$ *for brevity) if for some* $\omega \geq 0$ *and* $\forall x \in \mathbb{R}^d$

$$\mathbb{E}[\mathcal{C}(x)] = x, \qquad \mathbb{E}\left[\|\mathcal{C}(x) - x\|^2\right] \leq \omega \|x\|^2. \qquad (3)$$

A notable example of a mapping from this class is the *random sparsification* (Rand-q for $q \in \{1, \ldots, d\}$) operator defined by

$$\mathcal{C}_{\texttt{Rand-q}}(x) := \mathbf{C}_q x = \frac{d}{q} \sum_{i \in S} e_i e_i^\top x, \qquad (4)$$

where $e_1, \ldots, e_d \in \mathbb{R}^d$ are standard unit basis vectors in $\mathbb{R}^d$, and $S$ is a random subset of $[d] := \{1, \ldots, d\}$ sampled from the uniform distribution on the all subsets of $[d]$ with cardinality $q$. Rand-q belongs to $\mathbb{U}(d/q - 1)$, which means that the more elements are "dropped" (lower $q$), the higher is the variance $\omega$ of the compressor.

In this work, we are mainly interested in a somewhat more general class of operators than mere sparsifiers. In particular, we are interested in compressing via the application of random matrices, i.e., via *sketching*. A sketch $\mathbf{C}_i^k \in \mathbb{R}^{d \times d}$ can be used to represent submodel computations in the following way:

$$g_i^k := \mathbf{C}_i^k \nabla f_i(\mathbf{C}_i^k x^k), \qquad (5)$$

where we require $\mathbf{C}_i^k$ to be a symmetric positive semidefinite matrix. Such gradient estimate corresponds to computing the local gradient with respect to a sparse submodel model $\mathbf{C}_i^k x^k$, and additionally sketching the resulting gradient with the same matrix $\mathbf{C}_i^k$ to guarantee that the resulting update lies in the lower-dimensional subspace.

Using this notion, Algorithm 1 (with one local gradient step) can be represented in the following form

$$x^{k+1} = \frac{1}{n} \sum_{i=1}^{n} \left[ \mathbf{C}_i^k x^k - \gamma \mathbf{C}_i^k \nabla f_i(\mathbf{C}_i^k x^k) \right], \qquad (6)$$

which is equivalent to the SGD-type update (2) when **perfect reconstruction** property holds

$$\mathbf{C}^k := \frac{1}{n} \sum_{i=1}^{n} \mathbf{C}_i^k = \mathbf{I},$$

where $\mathbf{I}$ is the identity matrix (with probability one). This property holds for a specific class of compressors that are particularly useful for capturing the concept of an *independent* subnetwork partition.

**Definition 2** (Permutation sketch). *Assume that model size is greater than number of clients* $d \geq n$ *and* $d = qn$, *where* $q \geq 1$ *is an integer*[1]. *Let* $\pi = (\pi_1, \ldots, \pi_d)$ *be a random permutation of* $[d]$. *Then for all* $x \in \mathbb{R}^d$ *and each* $i \in [n]$ *we define* Perm-q *operator*

$$\mathbf{C}_i := n \cdot \sum_{j=q(i-1)+1}^{qi} e_{\pi_j} e_{\pi_j}^\top. \qquad (7)$$

---

[1] While this condition may look restrictive it naturally holds for distributed learning in a data-center setting. For other scenarios [40] generalized it for $n \geq d$ and block permutation case.

106 `Perm-q` is unbiased and can be conveniently used for representing (non-overlapping) structured
107 decomposition of the model such that every client $i$ is responsible for computations over a submodel
108 $\mathbf{C}_i x^k$.

109 Our convergence analysis relies on assumption previously used for coordinate descent type methods.

110 **Assumption 1** (Matrix smoothness). *A differentiable function* $f : \mathbb{R}^d \to \mathbb{R}$ *is* $\mathbf{L}$*-smooth, if there*
111 *exists a positive semi-definite matrix* $\mathbf{L} \in \mathbb{R}^{d \times d}$ *such that*

$$f(x + h) \leq f(x) + \langle \nabla f(x), h \rangle + \frac{1}{2} \langle \mathbf{L}h, h \rangle, \qquad \forall x, h \in \mathbb{R}^d. \tag{8}$$

112 Standard $L$-smoothness condition is obtained as a special case of (8) for $\mathbf{L} = L \cdot \mathbf{I}$.

### 2.1 Issues with existing approaches

114 Consider the simplest gradient type method with compressed model in the single node setting

$$x^{k+1} = x^k - \gamma \nabla f(\mathcal{C}(x^k)). \tag{9}$$

115 Algorithms belonging to this family require a different analysis in comparison to SGD [16, 18],
116 Distributed Compressed Gradient Descent [3, 26] and Randomized Coordinate Descent [34, 36] type
117 methods because the gradient estimator is no longer unbiased

$$\mathbb{E}\left[\nabla f(\mathcal{C}(x))\right] \neq \nabla f(x) = \mathbb{E}\left[\mathcal{C}(\nabla f(x))\right]. \tag{10}$$

118 That is why such kind of algorithms are harder to analyze. So, prior results for *unbiased* SGD [25]
119 can not be directly reused. Furthermore, the nature of the bias in this type of gradient estimator does
120 not exhibit additive (zero-mean) noise, thereby preventing the application of previous analyses for
121 biased SGD [1].

122 An assumption like bounded stochastic gradient norm extensively used in previous works [30, 48]
123 hinders an accurate understanding of such methods. This assumption hides the fundamental difficulty
124 of analyzing biased gradient estimator:

$$\mathbb{E}\left[\|\nabla f(\mathcal{C}(x))\|^2\right] \leq G \tag{11}$$

125 and may not hold even for quadratic functions $f(x) = x^\top \mathbf{A}x$. In addition, in the distributed
126 setting such condition can result in vacuous bounds [23] as it does not allow to accurately capture
127 heterogeneity.

## 3 Results in the Interpolation Case

129 To conduct a thorough theoretical analysis of methods that combine data with model parallelism,
130 we simplify the algorithm and problem setting to isolate the unique effects of this approach. The
131 following considerations are made:

132    (1) We assume that every node $i$ computes the true gradient at the submodel $\mathbf{C}_i \nabla f_i(\mathbf{C}_i x^k)$.

133    (2) A notable difference from the original IST algorithm 1 is that workers perform single
134        gradient descent step (or just gradient computation).

135    (3) Finally, we consider a special case of quadratic model (12) as a loss function (1).

136 Condition (1) is mainly for the sake of simplicity and clarity of exposition and can be potentially
137 generalized to stochastic gradient computations. (2) is imposed because local steps did not bring
138 any theoretical efficiency improvements for heterogeneous settings until very recently [32]. And
139 even then, only with the introduction of additional control variables, which goes against resource-
140 constrained device setting. The reason behind (3) is that despite the seeming simplicity quadratic
141 problem has been used extensively to study properties of neural networks [46, 49]. Moreover, it is a
142 non-trivial model which allows to understand complex optimization algorithms [4, 10, 17]. It serves
143 as a suitable problem for observing complex phenomena and providing theoretical insights, which
144 can also be observed in practical scenarios.

Having said that we consider a special case of problem (1)

$$f(x) = \frac{1}{n} \sum_{i=1}^{n} f_i(x), \qquad f_i(x) \equiv \frac{1}{2} x^\top \mathbf{L}_i x - \mathbf{b}_i^\top x. \tag{12}$$

In this case, $f(x)$ is $\overline{\mathbf{L}}$-smooth, and $\nabla f(x) = \overline{\mathbf{L}} x - \overline{\mathbf{b}}$, where $\overline{\mathbf{L}} = \frac{1}{n} \sum_{i=1}^{n} \mathbf{L}_i$ and $\overline{\mathbf{b}} := \frac{1}{n} \sum_{i=1}^{n} \mathbf{b}_i$.

## 3.1 No linear term: problems and solutions

First, let us examine the case of $\mathbf{b}_i \equiv 0$, which we call interpolation for quadratics, and perform the analysis for general sketches $\mathbf{C}_i^k$. In this case the gradient estimator (2) takes the form

$$g^k = \frac{1}{n} \sum_{i=1}^{n} \mathbf{C}_i^k \nabla f_i(\mathbf{C}_i^k x^k) = \frac{1}{n} \sum_{i=1}^{n} \mathbf{C}_i^k \mathbf{L}_i \mathbf{C}_i^k x^k = \overline{\mathbf{B}}^k x^k \tag{13}$$

where $\overline{\mathbf{B}}^k := \frac{1}{n} \sum_{i=1}^{n} \mathbf{C}_i^k \mathbf{L}_i \mathbf{C}_i^k$. We prove the following result for a method with such an estimator.

**Theorem 1.** *Consider the method* (2) *with estimator* (13) *for a quadratic problem* (12) *with* $\overline{\mathbf{L}} \succ 0$ *and* $\mathbf{b}_i \equiv 0$. *Then if* $\overline{\mathbf{W}} := \frac{1}{2} \mathbb{E} \left[ \overline{\mathbf{L}} \, \overline{\mathbf{B}}^k + \overline{\mathbf{B}}^k \overline{\mathbf{L}} \right] \succeq 0$ *and there exists constant* $\theta > 0$:

$$\mathbb{E} \left[ \overline{\mathbf{B}}^k \overline{\mathbf{L}} \, \overline{\mathbf{B}}^k \right] \preceq \theta \overline{\mathbf{W}}, \tag{14}$$

*and the step size is chosen as* $0 < \gamma \leq \frac{1}{\theta}$, *the iterates satisfy*

$$\frac{1}{K} \sum_{k=0}^{K-1} \mathbb{E} \left[ \left\| \nabla f(x^k) \right\|_{\overline{\mathbf{L}}^{-1} \overline{\mathbf{W}} \overline{\mathbf{L}}^{-1}}^2 \right] \leq \frac{2 \left( f(x^0) - \mathbb{E} \left[ f(x^K) \right] \right)}{\gamma K}, \tag{15}$$

*and*

$$\mathbb{E} \left[ \| x^k - x^\star \|_{\overline{\mathbf{L}}}^2 \right] \leq \left( 1 - \gamma \lambda_{\min} \left( \overline{\mathbf{L}}^{-\frac{1}{2}} \overline{\mathbf{W}} \overline{\mathbf{L}}^{-\frac{1}{2}} \right) \right)^k \| x^0 - x^\star \|_{\overline{\mathbf{L}}}^2. \tag{16}$$

This theorem establishes an $\mathcal{O}(1/K)$ convergence rate with constant step size up to a stationary point and linear convergence for the expected distance to the optimum. Note that we employ weighted norms in our analysis, as the considered class of loss functions satisfies the matrix $\overline{\mathbf{L}}$-smoothness Assumption 1. The use of standard Euclidean distance may result in loose bounds that do not recover correct rates for special cases like Gradient Descent.

It is important to highlight that inequality (14) may not hold (for any $\theta > 0$) in the general case as the matrix $\overline{\mathbf{W}}$ is not guaranteed to be positive (semi-)definite in the case of general sampling. The intuition behind it is that arbitrary sketches $\mathbf{C}_i^k$ can result in gradient estimator $g^k$, which is misaligned with the true gradient $\nabla f(x^k)$. Specifically, the inner product $\langle \nabla f(x^k), g^k \rangle$ can be negative, and there is no expected descent after one step.

Next, we give examples of samplings for which the inequality (14) can be satisfied.

**1. Identity.** Consider $\mathbf{C}_i \equiv \mathbf{I}$. Then $\overline{\mathbf{B}}^k = \overline{\mathbf{L}}$, $\overline{\mathbf{B}}^k \overline{\mathbf{L}} \, \overline{\mathbf{B}}^k = \overline{\mathbf{L}}^3$, $\overline{\mathbf{W}} = \overline{\mathbf{L}}^2 \succ 0$ and hence (14) is satisfied for $\theta = \lambda_{\max}(\overline{\mathbf{L}})$. So, (15) says that if we choose $\gamma = \frac{1}{\theta}$, then

$$\frac{1}{K} \sum_{k=0}^{K-1} \left\| \nabla f(x^k) \right\|_{\mathbf{I}}^2 \leq \frac{2 \lambda_{\max}(\overline{\mathbf{L}}) \left( f(x^0) - f(x^K) \right)}{K},$$

which exactly matches the rate of Gradient Descent in the non-convex setting. As for iterates convergence, the rate in (16) is $\lambda_{\max}(\overline{\mathbf{L}})/\lambda_{\min}(\overline{\mathbf{L}})$ corresponding to precise Gradient Descent result for strongly convex functions.

**2. Permutation.** Assume $n = d^2$ and the use of `Perm-1` (special case of Definition 2) sketch $\mathbf{C}_i^k = n e_{\pi_i^k} e_{\pi_i^k}^\top$, where $\pi^k = (\pi_1^k, \ldots, \pi_n^k)$ is a random permutation of $[n]$. Then

$$\mathbb{E} \left[ \overline{\mathbf{B}}^k \right] = \frac{1}{n} \sum_{i=1}^{n} n^2 \mathbb{E} \left[ \mathbf{C}_i^k \mathbf{L}_i \mathbf{C}_i^k \right] = \frac{1}{n} \sum_{i=1}^{n} n \mathrm{Diag}(\mathbf{L}_i) = \sum_{i=1}^{n} \mathbf{D}_i = n \overline{\mathbf{D}},$$

---

[2]This is done mainly for simplifying the presentation. Results can be generalized to the case of $n \neq d$ in the similar way as done in [40] which can be found in the Appendix.

where $\overline{\mathbf{D}} := \frac{1}{n} \sum_{i=1}^{n} \mathbf{D}_i, \mathbf{D}_i := \mathrm{Diag}(\mathbf{L}_i)$. Then inequality (14) leads to

$$n\,\overline{\mathbf{D}}\,\overline{\mathbf{L}}\,\overline{\mathbf{D}} \preceq \tfrac{\theta}{2}\left(\overline{\mathbf{L}}\,\overline{\mathbf{D}} + \overline{\mathbf{D}}\,\overline{\mathbf{L}}\right), \tag{17}$$

which may not always hold as $\overline{\mathbf{L}}\,\overline{\mathbf{D}} + \overline{\mathbf{D}}\,\overline{\mathbf{L}}$ is not guaranteed to be positive definite even in case of $\overline{\mathbf{L}} \succ 0$. However, such kind of condition can be enforced via a slight modification of permutation sketches $\{\tilde{\mathbf{C}}_i\}_{i=1}^{n}$, which is done in Section 3.1.2. The limitation of such an approach is that compressors $\tilde{\mathbf{C}}_i$ become no longer unbiased.

**Remark 1.** *Matrix $\overline{\mathbf{W}}$ in case of permutation sketches may not be positive-definite. Consider the following homogeneous ($\mathbf{L}_i \equiv \mathbf{L}$) two-dimensional problem example*

$$\mathbf{L} = \left[\begin{array}{cc} a & c \\ c & b \end{array}\right]. \tag{18}$$

*Then*

$$\overline{\mathbf{W}} = \tfrac{1}{2}\left[\overline{\mathbf{L}}\,\overline{\mathbf{D}} + \overline{\mathbf{D}}\,\overline{\mathbf{L}}\right] = \left[\begin{array}{cc} a^2 & c(a+b)/2 \\ c(a+b)/2 & b^2 \end{array}\right], \tag{19}$$

*which for $c > \frac{2ab}{a+b}$ has $\det(\overline{\mathbf{W}}) < 0$, and thus $\overline{\mathbf{W}} \not\succ 0$ according to Sylvester's criterion.*

Next, we focus on the particular case of **Permutation** sketches, which are the most suitable for model partitioning according to Independent Subnetwork Training (IST). At the rest of the section, we discuss how the condition (14) can be enforced via a specially designed preconditioning of the problem (12) or modification of sketch mechanism (7).

### 3.1.1 Homogeneous problem preconditioning

To start consider a homogeneous setting $f_i(x) = \frac{1}{2} x^\top \mathbf{L} x$, so $\mathbf{L}_i \equiv \mathbf{L}$. Now define $\mathbf{D} = \mathrm{Diag}(\mathbf{L})$ – diagonal matrix with elements equal to diagonal of $\mathbf{L}$. Then problem can be converted to

$$f_i(\mathbf{D}^{-\frac{1}{2}}x) = \tfrac{1}{2}\left(\mathbf{D}^{-\frac{1}{2}}x\right)^\top \mathbf{L}\left(\mathbf{D}^{-\frac{1}{2}}x\right) = \tfrac{1}{2}x^\top \underbrace{\left(\mathbf{D}^{-\frac{1}{2}}\mathbf{L}\mathbf{D}^{-\frac{1}{2}}\right)}_{\tilde{\mathbf{L}}} x, \tag{20}$$

which is equivalent to the original problem after a change of variables $\tilde{x} := \mathbf{D}^{-\frac{1}{2}}x$. Note that $\mathbf{D} = \mathrm{Diag}(\mathbf{L})$ is positive definite as $\mathbf{L} \succ 0$, and therefore $\tilde{\mathbf{L}} \succ 0$. Moreover, the preconditioned matrix $\tilde{\mathbf{L}}$ has all ones on the diagonal: $\mathrm{Diag}(\tilde{\mathbf{L}}) = \mathbf{I}$. If we now combine it with Perm-1 sketches

$$\mathbb{E}\left[\overline{\mathbf{B}}^k\right] = \mathbb{E}\left[\tfrac{1}{n}\sum_{i=1}^{n}\mathbf{C}_i\,\tilde{\mathbf{L}}\,\mathbf{C}_i\right] = n\mathrm{Diag}(\tilde{\mathbf{L}}) = n\mathbf{I}.$$

Therefore, inequality (14) takes the form $\tilde{\mathbf{W}} = n\tilde{\mathbf{L}} \succeq \frac{1}{\theta}n^2\tilde{\mathbf{L}}$, which holds for $\theta \geq n$, and left hand side of (15) can be transformed the following way

$$\left\|\nabla f(x^k)\right\|_{\tilde{\mathbf{L}}^{-1}\,\tilde{\mathbf{W}}\,\tilde{\mathbf{L}}^{-1}}^2 \geq n\lambda_{\min}\left(\tilde{\mathbf{L}}^{-1}\right)\left\|\nabla f(x^k)\right\|_{\mathbf{I}}^2 = n\lambda_{\max}(\tilde{\mathbf{L}})\left\|\nabla f(x^k)\right\|_{\mathbf{I}}^2 \tag{21}$$

for an accurate comparison to standard methods. The resulting convergence guarantee

$$\tfrac{1}{K}\sum_{k=0}^{K-1}\mathbb{E}\left[\left\|\nabla f(x^k)\right\|_{\mathbf{I}}^2\right] \leq \tfrac{2\lambda_{\max}(\tilde{\mathbf{L}})\left(f(x^0)-\mathbb{E}\left[f(x^K)\right]\right)}{K}, \tag{22}$$

which matches classical Gradient Descent.

### 3.1.2 Heterogeneous sketch preconditioning

In contrast to homogeneous case the heterogeneous problem $f_i(x) = \frac{1}{2}x^\top \mathbf{L}_i x$ can not be so easily preconditioned by a simple change of variables $\tilde{x} := \mathbf{D}^{-\frac{1}{2}}x$, as every client $i$ has its own matrix $\mathbf{L}_i$. However, this problem can be fixed via the following modification of Perm-1, which scales the output according to the diagonal elements of local smoothness matrix $\mathbf{L}_i$:

$$\tilde{\mathbf{C}}_i := \sqrt{n}\left[\mathbf{L}_i^{-\frac{1}{2}}\right]_{\pi_i,\pi_i} e_{\pi_i} e_{\pi_i}^\top. \tag{23}$$

In this case $\mathbb{E}\left[\tilde{\mathbf{C}}_i \mathbf{L}_i \tilde{\mathbf{C}}_i\right] = \mathbf{I}$, $\mathbb{E}\left[\overline{\mathbf{B}}^k\right] = \mathbf{I}$, and $\overline{\mathbf{W}} = \overline{\mathbf{L}}$. Then inequality (14) is satisfied for $\theta \geq 1$.

If one plugs these results into (15), such convergence guarantee can be obtained

$$\frac{1}{K}\sum_{k=0}^{K-1}\mathbb{E}\left[\left\|\nabla f(x^k)\right\|_{\mathbf{I}}^2\right] \leq \frac{2\lambda_{\max}(\overline{\mathbf{L}})\left(f(x^0)-\mathbb{E}\left[f(x^K)\right]\right)}{K}, \tag{24}$$

which matches the Gradient Descent result as well. Thus we can conclude that heterogeneity does not bring such a fundamental challenge in this scenario. In addition, a method with `Perm-1` is significantly better in terms of computational and communication complexity as it requires calculating the local gradients with respect to much smaller submodels and transmits only sparse updates.

This construction also shows that for $\gamma = 1/\theta = 1$

$$\gamma\lambda_{\min}\left(\overline{\mathbf{L}}^{-\frac{1}{2}}\overline{\mathbf{W}}\,\overline{\mathbf{L}}^{-\frac{1}{2}}\right) = \lambda_{\min}\left(\overline{\mathbf{L}}^{-\frac{1}{2}}\overline{\mathbf{L}}\,\overline{\mathbf{L}}^{-\frac{1}{2}}\right) = 1, \tag{25}$$

which after plugging into the bound for the iterates (16) shows that the method basically converges in 1 iteration. This observation that sketch preconditioning can be extremely efficient, although it uses only the diagonal elements of matrices $\mathbf{L}_i$.

Now when we understand that the method can perform very well in the special case of $\tilde{\mathbf{b}}_i \equiv 0$ we can move on to a more complicated situation.

## 4   Irreducible Bias in the General Case

Now we look at the most general heterogeneous case with different matrices and linear terms $f_i(x) \equiv \frac{1}{2}x^\top \mathbf{L}_i x - x^\top \mathbf{b}_i$. In this instance gradient estimator (2) takes the form

$$g^k = \frac{1}{n}\sum_{i=1}^n \mathbf{C}_i^k \nabla f_i(\mathbf{C}_i^k x^k) = \frac{1}{n}\sum_{i=1}^n \mathbf{C}_i^k\left(\mathbf{L}_i \mathbf{C}_i^k x^k - \mathbf{b}_i\right) = \overline{\mathbf{B}}^k x^k - \overline{\mathbf{Cb}}, \tag{26}$$

where $\overline{\mathbf{Cb}} = \frac{1}{n}\sum_{i=1}^n \mathbf{C}_i^k\, \mathbf{b}_i$. Herewith let us use a heterogeneous permutation sketch preconditioner (23) like in Section 3.1.2 Then $\mathbb{E}\left[\overline{\mathbf{B}}^k\right] = \mathbf{I}$ and $\mathbb{E}\left[\overline{\mathbf{Cb}}\right] = \frac{1}{\sqrt{n}}\widetilde{\mathbf{D}\,\mathbf{b}}$, where $\widetilde{\mathbf{D}\,\mathbf{b}} := \frac{1}{n}\sum_{i=1}^n \mathbf{D}_i^{-\frac{1}{2}}\, \mathbf{b}_i$. Furthermore expected gradient estimator (26) results in $\mathbb{E}\left[g^k\right] = x^k - \frac{1}{\sqrt{n}}\widetilde{\mathbf{D}\,\mathbf{b}}$ and can be transformed the following way

$$\mathbb{E}\left[g^k\right] = \overline{\mathbf{L}}^{-1}\overline{\mathbf{L}}\,x^k \pm \overline{\mathbf{L}}^{-1}\overline{\mathbf{b}} - \frac{1}{\sqrt{n}}\widetilde{\mathbf{D}\,\mathbf{b}} = \overline{\mathbf{L}}^{-1}\nabla f(x^k) + \underbrace{\overline{\mathbf{L}}^{-1}\overline{\mathbf{b}} - \frac{1}{\sqrt{n}}\widetilde{\mathbf{D}\,\mathbf{b}}}_{h}, \tag{27}$$

which reflects the decomposition of the estimator into optimally preconditioned true gradient and a bias, depending on the linear terms $\mathbf{b}_i$.

### 4.1   Bias of the method

Estimator (27) can be directly plugged (with proper conditioning) into general SGD update (2)

$$\mathbb{E}\left[x^{k+1}\right] = x^k - \gamma\mathbb{E}\left[g^k\right] = (1-\gamma)x^k + \frac{\gamma}{\sqrt{n}}\widetilde{\mathbf{D}\,\mathbf{b}} = (1-\gamma)^{k+1}x^0 + \frac{\gamma}{\sqrt{n}}\widetilde{\mathbf{D}\,\mathbf{b}}\sum_{j=0}^k (1-\gamma)^j. \tag{28}$$

The resulting recursion (28) is exact, and its asymptotic limit can be analyzed. Thus for constant $\gamma < 1$ by using the formula for the sum of the first $k$ terms of a geometric series, one gets

$$\mathbb{E}\left[x^k\right] = (1-\gamma)^k x^0 + \frac{1-(1-\gamma)^k}{\sqrt{n}}\widetilde{\mathbf{D}\,\mathbf{b}} \xrightarrow[k\to\infty]{} \frac{1}{\sqrt{n}}\widetilde{\mathbf{D}\,\mathbf{b}},$$

which shows that in the limit, the first initialization term (with $x^0$) vanishes while the second converges to $\frac{1}{\sqrt{n}}\widetilde{\mathbf{D}\,\mathbf{b}}$. This reasoning shows that the method does not converge to the exact solution

$$x^k \to x^\infty \neq x^\star \in \arg\min_{x\in\mathbb{R}^d}\left\{\tfrac{1}{2}x^\top \overline{\mathbf{L}}\,x - x^\top \overline{\mathbf{b}}\right\},$$

which for the positive-definite $\overline{\mathbf{L}}$ can be defined as $x^\star = \overline{\mathbf{L}}^{-1}\overline{\mathbf{b}}$, while $x^\infty = \frac{1}{n\sqrt{n}}\sum_{i=1}^n \mathbf{D}_i^{-\frac{1}{2}}\, \mathbf{b}_i$. So, in general, there is an unavoidable bias. However, in the limit case: $n = d \to \infty$, the bias diminishes.

## 4.2 Generic convergence analysis

While the analysis in Section 4.1 is precise, it does not allow us to compare the convergence of IST to standard optimization methods. Due to this, we also analyze the non-asymptotic behavior of the method to understand the convergence speed. Our result is formalized in the following theorem.

**Theorem 2.** *Consider the method* (2) *with estimator* (26) *for a quadratic problem* (12) *with the positive definite matrix* $\overline{\mathbf{L}} \succ 0$. *Assume that for every* $\mathbf{D}_i := \mathrm{Diag}(\mathbf{L}_i)$ *matrices* $\mathbf{D}_i^{-\frac{1}{2}}$ *exist, scaled permutation sketches* (23) *are used and heterogeneity is bounded as* $\mathbb{E}\left[\left\|g^k - \mathbb{E}\left[g^k\right]\right\|_{\overline{\mathbf{L}}}^2\right] \leq \sigma^2$. *Then for step size is chosen as*

$$0 < \gamma \leq \gamma_{c,\beta} := \tfrac{1/2 - \beta}{\beta + 1/2}, \tag{29}$$

*where* $\gamma_{c,\beta} \in (0, 1]$ *for* $\beta \in (0, 1/2)$, *the iterates satisfy*

$$\frac{1}{K}\sum_{k=0}^{K-1} \mathbb{E}\left[\left\|\nabla f(x^k)\right\|_{\overline{\mathbf{L}}^{-1}}^2\right] \leq \frac{2\left(f(x^0) - \mathbb{E}[f(x^K)]\right)}{\gamma K} + \left(2\beta^{-1}\left(1 - \gamma\right) + \gamma\right)\|h\|_{\overline{\mathbf{L}}}^2 + \gamma\sigma^2, \tag{30}$$

*where* $\overline{\mathbf{L}} = \frac{1}{n}\sum_{i=1}^n \mathbf{L}_i, h = \overline{\mathbf{L}}^{-1}\overline{\mathrm{b}} - \frac{1}{\sqrt{n}}\frac{1}{n}\sum_{i=1}^n \mathbf{D}_i^{-\frac{1}{2}}\mathrm{b}_i$ *and* $\overline{\mathrm{b}} = \frac{1}{n}\sum_{i=1}^n \mathrm{b}_i$.

Note that the derived convergence upper bound has a neighborhood proportional to the bias of the gradient estimator $h$ and level of heterogeneity $\sigma^2$. Some of these terms with factor $\gamma$ can be eliminated via decreasing learning rate schedule (e.g., $\sim 1/\sqrt{k}$). However, such a strategy does not diminish the term with a multiplier $2\beta^{-1}(1 - \gamma)$, making the neighborhood irreducible. Moreover, this term can be eliminated for $\gamma = 1$, which also minimizes the first term that decreases as $1/K$. Though, such step size choice maximizes the terms with factor $\gamma$. Furthermore, there exists an inherent trade-off between convergence speed and the size of the neighborhood.

In addition, convergence to the stationary point is measured in the weighted by $\overline{\mathbf{L}}^{-1}$ squared norm of the gradient. At the same time, the neighborhood term depends on the weighted by $\overline{\mathbf{L}}$ norm of $h$. This fine-grained decoupling is achieved by carefully applying Fenchel-Young inequality and provides a tighter characterization of the convergence compared to using standard Euclidean distances.

**Homogeneous case.** In this scenario, every worker has access to the all data $f_i(x) \equiv \frac{1}{2}x^\top \mathbf{L}x - x^\top \mathrm{b}$. Then diagonal preconditioning of the problem can be used as in the previous Section 3.1.1. This results in a gradient $\nabla f(x) = \tilde{\mathbf{L}}x - \tilde{\mathrm{b}}$ for $\tilde{\mathbf{L}} = \mathbf{D}^{-\frac{1}{2}}\mathbf{L}\mathbf{D}^{-\frac{1}{2}}$ and $\tilde{\mathrm{b}} = \mathbf{D}^{-\frac{1}{2}}\mathrm{b}$. If it is further combined with a scaled by $1/\sqrt{n}$ Permutation sketch $\mathbf{C}_i := \sqrt{n}e_{\pi_i}e_{\pi_i}^\top$, the resulting gradient estimator is

$$g^k = x^k - \frac{1}{\sqrt{n}}\tilde{\mathrm{b}} = \tilde{\mathbf{L}}^{-1}\nabla f(x^k) + \tilde{h}, \tag{31}$$

for $\tilde{h} = \tilde{\mathbf{L}}^{-1}\tilde{\mathrm{b}} - \frac{1}{\sqrt{n}}\tilde{\mathrm{b}}$. In this case heterogeneity term $\sigma^2$ from upper bound (30) disappears as $\mathbb{E}\left[\left\|g^k - \mathbb{E}\left[g^k\right]\right\|_{\overline{\mathbf{L}}}^2\right] = 0$, thus the neighborhood size can significantly decrease. However, the bias term depending on $\tilde{h}$ still remains as the method does not converge to the exact solution $x^k \to x^\infty \neq x^\star = \tilde{\mathbf{L}}^{-1}\tilde{\mathrm{b}}$ for positive-definite $\tilde{\mathbf{L}}$. Nevertheless the method's fixed point $x^\infty = \tilde{\mathrm{b}}/\sqrt{n}$ and solution $x^\star$ can coincide when $\tilde{\mathbf{L}}^{-1}\tilde{\mathrm{b}} = \frac{1}{\sqrt{n}}\tilde{\mathrm{b}}$, which means that $\tilde{\mathrm{b}}$ is the right eigenvector of matrix $\tilde{\mathbf{L}}^{-1}$ with eigenvalue $\frac{1}{\sqrt{n}}$.

Let us contrast obtained result (30) with non-convex rate of SGD [25] with constant step size $\gamma$ for $L$-smooth and lower-bounded $f$

$$\min_{k \in \{0, \dots, K-1\}}\left\|\nabla f(x^k)\right\|^2 \leq \frac{6\left(f(x^0) - \inf f\right)}{\gamma K} + \gamma LC, \tag{32}$$

where constant $C$ depends, for example, on the variance of stochastic gradient estimates. Observe that the first term in the compared upper bounds (32) and (30) is almost identical and decreases with speed $1/K$. But unlike (30) the neighborhood for SGD can be completely eliminated by reducing the step size $\gamma$. This highlights a fundamental difference of our results to unbiased methods.

The intuition behind this issue is that for SGD-type methods like Compressed Gradient Descent

$$x^{k+1} = x^k - \mathcal{C}(\nabla f(x^k)) \tag{33}$$

the gradient estimate is unbiased and enjoys the property that variance

$$\mathbb{E}\left[\|\mathcal{C}(\nabla f(x^k)) - \nabla f(x^k)\|^2\right] \leq \omega \|\nabla f(x^k)\|^2 \tag{34}$$

goes down to zero as the method progresses because $\nabla f(x^k) \to \nabla f(x^\star) = 0$ in the unconstrained case. In addition, any stationary point $x^\star$ ceases to be a fixed point of the iterative procedure as

$$x^\star \neq x^\star - \nabla f(\mathcal{C}(x^\star)), \tag{35}$$

in the general case, unlike for Compressed Gradient Descent with both biased and unbiased compressors $\mathcal{C}$. So, even if the method (computing gradient at sparse model) is initialized from the *solution* after one gradient step, it may get away from there.

## 4.3 Comparison to previous works

**Independent Subnetwork Training [45].**    There are several improvements over the previous works that tried to theoretically analyze the convergence of Distributed IST.

The first difference is that our results allow for an almost arbitrary level of model sparsification, i.e., work for any $\omega \geq 0$ as permutation sketches can be viewed as a special case of compression operators (1). This improves significantly over the work of [45], which demands[3] $\omega \lesssim \mu^2/L^2$. Such a requirement is very restrictive as the condition number $L/\mu$ of the loss function $f$ is typically very large for any non-trivial optimization problem. Thus, the sparsifier's (4) variance $\omega = d/q - 1$ has to be very close to 0 and $q \approx d$. So, the previous theory allows almost no compression (sparsification) because it is based on the analysis of Gradient Descent with Compressed Iterates [24].

The second distinction is that the original IST work [45] considered a single node setting and thus their convergence bounds did not capture the effect of heterogeneity, which we believe is of crucial importance for distributed setting [9, 39]. Besides, they consider Lipschitz continuity of the loss function $f$, which is not satisfied for a simple quadratic model. A more detailed comparison including additional assumptions on the gradient estimator made in [45] is presented in the Appendix.

**FL with Model Pruning.**    In a recent work [48] made an attempt to analyze a variant of the FedAvg algorithm with sparse local initialization and compressed gradient training (pruned local models). They considered a case of $L$-smooth loss and sparsification operator satisfying a similar condition to (1). However, they also assumed that the squared norm of stochastic gradient is uniformly bounded (11), which is "pathological" [23] especially in the case of local methods as it does not allow to capture the very important effect of heterogeneity and can result in vacuous bounds.

In the Appendix we show some limitations of other relevant previous approaches to training with compressed models: too restrictive assumptions on the algorithm [33] or not applicability in our problem setting [7].

## 5    Conclusions and Future Work

In this study, we introduced a novel approach to understanding training with combined model and data parallelism for a quadratic model. This framework allowed to shed light on distributed submodel optimization which revealed the advantages and limitations Independent Subnetwork Training (IST). Moreover, we accurately characterized the behavior of the considered method in both homogeneous and heterogeneous scenarios without imposing restrictive assumptions on gradient estimators.

In future research, it would be valuable to explore extensions of our findings to settings that are closer to practical scenarios, such as cross-device federated learning. This could involve investigating partial participation support, leveraging local training benefits, and ensuring robustness against stragglers. Additionally, it would be interesting to generalize our results to non-quadratic scenarios without relying on pathological assumptions.

---

[3] $\mu$ refers to constant from Polyak-Łojasiewicz (or strong convexity) condition. In case of a quadratic problem with positive-definite matrix $\mathbf{A}$: $\mu = \lambda_{\min}(\mathbf{A})$

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
