**Basic Inequalities.** For all vectors $a, b \in \mathbb{R}^d$ and random vector $X \in \mathbb{R}^d$:

$$2 \langle a, b \rangle = \|a\|^2 + \|b\|^2 - \|a - b\|^2, \tag{37}$$

$$\mathbf{E} \|X - a\|^2 = \mathbf{E} \|X - \mathbf{E} X\|^2 + \| \mathbf{E} X - a\|^2. \tag{38}$$

**Lemma 1** (Fenchel–Young inequality). *For any function $f$ and its convex conjugate $f^*$, Fenchel's inequality (also known as the Fenchel–Young inequality) holds for every $x, y \in \mathbb{R}^d$*

$$\langle x, y \rangle \leq f(x) + f^*(y).$$

*The proof follows from the definition of conjugate: $f^*(y) := \sup_{x'} \{ \langle y, x' \rangle - f(x') \} \geq \langle y, x \rangle - f(x)$.*

In the case of a quadratic function $f(x) = \beta \|x\|_{\mathbf{L}}^2$ we can compute $f^*(y) = \frac{1}{4} \beta^{-1} \|y\|_{\mathbf{L}^{-1}}^2$. Thus

$$\langle x, y \rangle \leq \beta \|x\|_{\mathbf{L}}^2 + \frac{1}{4} \beta^{-1} \|y\|_{\mathbf{L}^{-1}}^2. \tag{39}$$

## B  Proofs

### B.1  Permutation sketch computations

All derivations in this section are performed for $n = d$ case.

**Classical Permutation Sketching.** `Perm-1`: $\mathbf{C}_i = n e_{\pi_i} e_{\pi_i}^\top$, where $\pi = (\pi_1, \dots, \pi_n)$ is a random permutation of $[n]$. For homogeneous problem $\mathbf{L}_i \equiv \mathbf{L}$:

$$\mathbb{E} \left[ \overline{\mathbf{B}}^k \right] = \mathbb{E} \left[ \frac{1}{n} \sum_{i=1}^n \mathbf{C}_i \, \mathbf{L} \, \mathbf{C}_i \right] = n \mathrm{Diag}(\mathbf{L}) \tag{40}$$

Then

$$2 \, \overline{\mathbf{W}} = \mathbb{E} \left[ \mathbf{L} \, \overline{\mathbf{B}}^k + \overline{\mathbf{B}}^k \, \mathbf{L} \right] = n \left( \mathbf{L} \mathrm{Diag}(\mathbf{L}) + \mathrm{Diag}(\mathbf{L}) \mathbf{L} \right) \tag{41}$$

and

$$\mathbb{E} \left[ \overline{\mathbf{B}}^k \, \mathbf{L} \, \overline{\mathbf{B}}^k \right] = n^2 \mathrm{Diag}(\mathbf{L}) \mathbf{L} \mathrm{Diag}(\mathbf{L}). \tag{42}$$

Almost the same calculations can be performed for $\mathbf{C}_i' = \sqrt{n} e_{\pi_i} e_{\pi_i}^\top$.

#### B.1.1  Heterogeneous sketch preconditioning.

We recall the following modification of `Perm-1`:

$$\tilde{\mathbf{C}}_i := \sqrt{n} \left[ \mathbf{L}_i^{-\frac{1}{2}} \right]_{\pi_i, \pi_i} e_{\pi_i} e_{\pi_i}^\top. \tag{43}$$

Then

$$\mathbb{E} \left[ \tilde{\mathbf{C}}_i \mathbf{L}_i \tilde{\mathbf{C}}_i \right] = \mathbb{E} \left[ n [\mathbf{L}_i^{-1}]_{\pi_i, \pi_i} e_{\pi_i} e_{\pi_i}^\top \mathbf{L}_i e_{\pi_i} e_{\pi_i}^\top \right] = \frac{1}{n} \sum_{j=1}^n n e_j \mathbf{I}_{j,j} e_j^\top = \mathbf{I}. \tag{44}$$

513 and

$$\mathbb{E}\left[\overline{\mathbf{B}}^k\right] = \mathbb{E}\left[\frac{1}{n}\sum_{i=1}^{n}\tilde{\mathbf{C}}_i\mathbf{L}_i\tilde{\mathbf{C}}_i\right]$$

$$= \frac{1}{n}\sum_{i=1}^{n}\mathbb{E}\left[n[\mathbf{L}_i^{-1}]_{\pi_i,\pi_i}e_{\pi_i}e_{\pi_i}^\top\mathbf{L}_ie_{\pi_i}e_{\pi_i}^\top\right]$$

$$= \frac{1}{n}\sum_{i=1}^{n}\frac{1}{n}\sum_{j=1}^{n}n[\mathbf{L}_i^{-1}]_{j,j}e_j[\mathbf{L}_i]_{jj}e_j^\top$$

$$= \frac{1}{n}\sum_{i=1}^{n}\sum_{j=1}^{n}e_je_j^\top$$

$$= \mathbf{I}.$$

514 Thus $\overline{\mathbf{W}} = \frac{1}{2}\mathbb{E}\left[\overline{\mathbf{L}}\,\overline{\mathbf{B}}^k + \overline{\mathbf{B}}^k\,\overline{\mathbf{L}}\right] = \overline{\mathbf{L}}$. For the left hand side of inequality (14) we have

$$\mathbb{E}\left[\overline{\mathbf{B}}^k\,\overline{\mathbf{L}}\,\overline{\mathbf{B}}^k\right] = \mathbb{E}\left[\frac{1}{n}\sum_{i=1}^{n}\tilde{\mathbf{C}}_i\mathbf{L}_i\tilde{\mathbf{C}}_i\,\overline{\mathbf{L}}\,\frac{1}{n}\sum_{i=j}^{n}\tilde{\mathbf{C}}_j\mathbf{L}_j\tilde{\mathbf{C}}_j\right]$$

$$= \frac{1}{n^2}\sum_{i,j=1}^{n}\mathbb{E}\left[\tilde{\mathbf{C}}_i\mathbf{L}_i\tilde{\mathbf{C}}_i\,\overline{\mathbf{L}}\,\tilde{\mathbf{C}}_j\mathbf{L}_j\tilde{\mathbf{C}}_j\right]$$

$$= \sum_{i,j=1}^{n}e_ie_i^\top\,\overline{\mathbf{L}}\,e_je_j^\top$$

$$= \mathbf{I}\,\overline{\mathbf{L}}\,\mathbf{I}$$

$$= \overline{\mathbf{L}}.$$

515 ## B.2  Interpolation case: proof of Theorem 1

516 In the quadratic interpolation regime the linear term is zero $f_i(x) = \frac{1}{2}x^\top\mathbf{L}_ix$, and gradient estimator
517 has the form

$$g^k = \frac{1}{n}\sum_{i=1}^{n}\mathbf{C}_i^k\nabla f_i(\mathbf{C}_i^kx^k) = \frac{1}{n}\sum_{i=1}^{n}\mathbf{C}_i^k\mathbf{L}_i\mathbf{C}_i^kx^k = \overline{\mathbf{B}}^k\,x^k. \qquad (45)$$

518 *Proof.* At first we prove **stationary point** convergence result (15).

519 Using $\overline{\mathbf{L}}$-smoothness of function $f$, we get

$$f(x^{k+1}) \overset{(2)}{=} f(x^k - \gamma g^k) \overset{(8)}{\leq} f(x^k) - \left\langle\nabla f(x^k), \gamma g^k\right\rangle + \frac{\gamma^2}{2}\left\|g^k\right\|_{\overline{\mathbf{L}}}^2$$

$$\overset{(13)}{=} f(x^k) - \gamma\left\langle\overline{\mathbf{L}}\,x^k, \overline{\mathbf{B}}^k\,x^k\right\rangle + \frac{\gamma^2}{2}\left\|\overline{\mathbf{B}}^k\,x^k\right\|_{\overline{\mathbf{L}}}^2$$

$$= f(x^k) - \gamma(x^k)^\top\,\overline{\mathbf{L}}\,\overline{\mathbf{B}}^k\,x^k + \frac{\gamma^2}{2}(x^k)^\top\,\overline{\mathbf{B}}^k\,\overline{\mathbf{L}}\,\overline{\mathbf{B}}^k\,x^k.$$

After applying conditional expectation, using its linearity, and the fact that

$$x^\top\mathbf{A}x = \frac{1}{2}x^\top\left(\mathbf{A} + \mathbf{A}^\top\right)x$$

 we get

$$
\begin{aligned}
\mathbb{E}\left[f(x^{k+1}) \mid x^k\right] \quad \leq \quad & f(x^k) - \gamma(x^k)^\top \mathbb{E}\left[\overline{\mathbf{L}}\,\overline{\mathbf{B}}^k\right] x^k + \frac{\gamma^2}{2}(x^k)^\top \mathbb{E}\left[\overline{\mathbf{B}}^k\,\overline{\mathbf{L}}\,\overline{\mathbf{B}}^k\right] x^k \\
= \quad & f(x^k) - \gamma(x^k)^\top \overline{\mathbf{W}}\, x^k + \frac{\gamma^2}{2}(x^k)^\top \mathbb{E}\left[\overline{\mathbf{B}}^k\,\overline{\mathbf{L}}\,\overline{\mathbf{B}}^k\right] x^k \\
= \quad & f(x^k) - \gamma(\nabla f(x^k))^\top \overline{\mathbf{L}}^{-1}\,\overline{\mathbf{W}}\,\overline{\mathbf{L}}^{-1}\, \nabla f(x^k) \\
& + \frac{\gamma^2}{2}(\nabla f(x^k))^\top \overline{\mathbf{L}}^{-1}\, \mathbb{E}\left[\overline{\mathbf{B}}^k\,\overline{\mathbf{L}}\,\overline{\mathbf{B}}^k\right] \overline{\mathbf{L}}^{-1}\, \nabla f(x^k) \\
\overset{(14)}{\leq} \quad & f(x^k) - \gamma\|\nabla f(x^k)\|^2_{\overline{\mathbf{L}}^{-1}\,\overline{\mathbf{W}}\,\overline{\mathbf{L}}^{-1}} + \frac{\theta\gamma^2}{2}\|\nabla f(x^k)\|^2_{\overline{\mathbf{L}}^{-1}\,\overline{\mathbf{W}}\,\overline{\mathbf{L}}^{-1}} \\
= \quad & f(x^k) - \gamma\left(1 - \theta\gamma/2\right)\|\nabla f(x^k)\|^2_{\overline{\mathbf{L}}^{-1}\,\overline{\mathbf{W}}\,\overline{\mathbf{L}}^{-1}} \\
\leq \quad & f(x^k) - \frac{\gamma}{2}\|\nabla f(x^k)\|^2_{\overline{\mathbf{L}}^{-1}\,\overline{\mathbf{W}}\,\overline{\mathbf{L}}^{-1}},
\end{aligned}
$$

where the last inequality holds for the stepsize $\gamma \leq \frac{1}{\theta}$.

Rearranging gives

$$
\left\|\nabla f(x^k)\right\|^2_{\overline{\mathbf{L}}^{-1}\,\overline{\mathbf{W}}\,\overline{\mathbf{L}}^{-1}} \leq \frac{2}{\gamma}\left(f(x^k) - \mathbb{E}\left[f(x^{k+1}) \mid x^k\right]\right),
$$

which after averaging gives the desired result

$$
\frac{1}{K}\sum_{k=0}^{K-1} \mathbb{E}\left[\left\|\nabla f(x^k)\right\|^2_{\overline{\mathbf{L}}^{-1}\,\overline{\mathbf{W}}\,\overline{\mathbf{L}}^{-1}}\right] \leq \frac{2}{\gamma K}\sum_{k=0}^{K-1}\left(f(x^k) - \mathbb{E}\left[f(x^{k+1})\right]\right) = \frac{2\left(f(x^0) - \mathbb{E}\left[f(x^K)\right]\right)}{\gamma K}.
$$
(46)

Now we show the result for the **iterates convergence** (16).

Expectation conditioned on $x^k$:

$$
\begin{aligned}
r^{k+1} \quad := \quad & \mathbb{E}\left[\|x^{k+1} - x^\star\|^2_{\overline{\mathbf{L}}}\right] \\
= \quad & \mathbb{E}\left[\|x^k - \gamma g^k - x^\star\|^2_{\overline{\mathbf{L}}}\right] \\
= \quad & \|x^k - x^\star\|^2_{\overline{\mathbf{L}}} - 2\gamma\left\langle x^k - x^\star, \mathbb{E}\left[\overline{\mathbf{L}}\,\overline{\mathbf{B}}^k\right](x^k - x^\star)\right\rangle + \gamma^2\left\langle \mathbb{E}\left[\overline{\mathbf{B}}^k\,\overline{\mathbf{L}}\,\overline{\mathbf{B}}^k\right](x^k - x^\star), x^k - x^\star\right\rangle \\
\overset{x^\star=0}{=} \quad & r^k - 2\gamma\left\langle x^k - x^\star, \overline{\mathbf{W}}(x^k - x^\star)\right\rangle + \gamma^2\left\langle x^k - x^\star, \mathbb{E}\left[\overline{\mathbf{B}}^k\,\overline{\mathbf{L}}\,\overline{\mathbf{B}}^k\right](x^k - x^\star)\right\rangle \\
\overset{(16)}{\leq} \quad & r^k - 2\gamma\left\langle x^k - x^\star, \overline{\mathbf{W}}(x^k - x^\star)\right\rangle + \theta\gamma^2\left\langle x^k - x^\star, \overline{\mathbf{W}}(x^k - x^\star)\right\rangle \\
= \quad & \|x^k - x^\star\|^2_{\overline{\mathbf{L}}} - 2\gamma\left(1 - \theta\gamma/2\right)\left\|\overline{\mathbf{L}}^{\frac{1}{2}}(x^k - x^\star)\right\|^2_{\overline{\mathbf{L}}^{-\frac{1}{2}}\,\overline{\mathbf{W}}\,\overline{\mathbf{L}}^{-\frac{1}{2}}} \\
\overset{\gamma\leq 1/\theta}{\leq} \quad & \|x^k - x^\star\|^2_{\overline{\mathbf{L}}} - \gamma\left\|\overline{\mathbf{L}}^{\frac{1}{2}}(x^k - x^\star)\right\|^2_{\overline{\mathbf{L}}^{-\frac{1}{2}}\,\overline{\mathbf{W}}\,\overline{\mathbf{L}}^{-\frac{1}{2}}} \\
\leq \quad & \|x^k - x^\star\|^2_{\overline{\mathbf{L}}} - \gamma\lambda_{\min}\left(\overline{\mathbf{L}}^{-\frac{1}{2}}\,\overline{\mathbf{W}}\,\overline{\mathbf{L}}^{-\frac{1}{2}}\right)\left\|\overline{\mathbf{L}}^{\frac{1}{2}}(x^k - x^\star)\right\|^2 \\
= \quad & \left(1 - \gamma\lambda_{\min}\left(\overline{\mathbf{L}}^{-\frac{1}{2}}\,\overline{\mathbf{W}}\,\overline{\mathbf{L}}^{-\frac{1}{2}}\right)\right)\|x^k - x^\star\|^2_{\overline{\mathbf{L}}}.
\end{aligned}
$$

After unrolling the recursion we obtain the convergence result

$$
\mathbb{E}\left[\|x^{k+1} - x^\star\|^2_{\overline{\mathbf{L}}}\right] \leq \left(1 - \gamma\lambda_{\min}\left(\overline{\mathbf{L}}^{-\frac{1}{2}}\,\overline{\mathbf{W}}\,\overline{\mathbf{L}}^{-\frac{1}{2}}\right)\right)^{k+1}\|x^0 - x^\star\|^2_{\overline{\mathbf{L}}}.
$$

$\square$

## B.3   Non-zero solution

For reminder in the most general case the problem has the form

$$
f(x) = \frac{1}{n}\sum_{i=1}^n f_i(x), \qquad f_i(x) \equiv \frac{1}{2}x^\top \mathbf{L}_i x - x^\top \mathrm{b}_i.
$$

with gradient estimator

$$g^k = \frac{1}{n} \sum_{i=1}^{n} \mathbf{C}_i^k \nabla f_i(\mathbf{C}_i^k x^k) = \frac{1}{n} \sum_{i=1}^{n} \mathbf{C}_i^k \left( \mathbf{L}_i \mathbf{C}_i^k x^k - \mathbf{b}_i \right) = \overline{\mathbf{B}}^k x^k - \frac{1}{n} \sum_{i=1}^{n} \mathbf{C}_i^k \mathbf{b}_i. \qquad (47)$$

**General calculations for estimator** (26). In the heterogeneous case the following sketch precondi-tioner is used

$$\tilde{\mathbf{C}}_i := \sqrt{n} \left[ \mathbf{L}_i^{-\frac{1}{2}} \right]_{\pi_i, \pi_i} e_{\pi_i} e_{\pi_i}^\top.$$

Then $\mathbb{E}\left[ \overline{\mathbf{B}}^k \right] = \mathbf{I}$ (calculation was done in Section B.1.1) and

$$
\begin{aligned}
\mathbb{E}\left[ \overline{\mathbf{Cb}} \right] &= \frac{1}{n} \sum_{i=1}^{n} \mathbb{E}\left[ \tilde{\mathbf{C}}_i^k \mathbf{b}_i \right] \\
&= \frac{1}{n} \sum_{i=1}^{n} \mathbb{E}\left[ \sqrt{n} [\mathbf{L}_i^{-\frac{1}{2}}]_{\pi_i, \pi_i} e_{\pi_i} e_{\pi_i}^\top \mathbf{b}_i \right] \\
&= \frac{1}{n} \sum_{i=1}^{n} \frac{1}{n} \sum_{j=1}^{n} \sqrt{n} [\mathbf{L}_i^{-\frac{1}{2}}]_{j,j} e_j [\mathbf{b}_i]_j \\
&= \frac{1}{n} \sum_{i=1}^{n} \frac{1}{n} \sqrt{n} \mathbf{D}_i^{-\frac{1}{2}} \mathbf{b}_i \\
&= \frac{1}{\sqrt{n}} \frac{1}{n} \sum_{i=1}^{n} \mathbf{D}_i^{-\frac{1}{2}} \mathbf{b}_i \\
&= \frac{1}{\sqrt{n}} \underbrace{\overline{\mathbf{D}^{-\frac{1}{2}} \mathbf{b}}}_{\widetilde{\mathbf{D} \mathbf{b}}}
\end{aligned}
$$

### B.3.1  Generic convergence analysis for heterogeneous case: proof of Theorem 2.

Here we formulate and prove a bit more general version of Theorem 2, which is obtained as a special case of the next result for $c = 1/2$.

**Theorem 3.** *Consider the method* (2) *with estimator* (26) *for a quadratic problem* (12) *with pos-itive definite matrix* $\overline{\mathbf{L}} \succ 0$. *Then if for every* $\mathbf{D}_i := \mathrm{Diag}(\mathbf{L}_i)$ *matrices* $\mathbf{D}_i^{-\frac{1}{2}}$ *exist, scaled permutation sketches* $\mathbf{C}_i := \sqrt{n}[\mathbf{L}_i^{-\frac{1}{2}}]_{\pi_i, \pi_i} e_{\pi_i} e_{\pi_i}^\top$ *are used and heterogeneity is bounded as* $\mathbb{E}\left[ \|g^k - \mathbb{E}\left[ g^k \right]\|_{\overline{\mathbf{L}}}^2 \right] \le \sigma^2$. *Then for step size is chosen as*

$$0 < \gamma \le \gamma_{c,\beta} := \frac{1 - c - \beta}{\beta + 1/2}, \qquad (48)$$

*where* $\gamma_{c,\beta} \in (0, 1]$ *for* $\beta + c < 1$, *the iterates satisfy*

$$\frac{1}{K} \sum_{k=0}^{K-1} \mathbb{E}\left[ \|\nabla f(x^k)\|_{\overline{\mathbf{L}}^{-1}}^2 \right] \le \frac{f(x^0) - \mathbb{E}\left[ f(x^K) \right]}{c\gamma K} + \left( \frac{1-\gamma}{c\beta} + \frac{\gamma}{2c} \right) \|h\|_{\overline{\mathbf{L}}}^2 + \frac{\gamma}{2c} \sigma^2. \qquad (49)$$

*where* $\overline{\mathbf{L}} = \frac{1}{n} \sum_{i=1}^{n} \mathbf{L}_i$, $h = \overline{\mathbf{L}}^{-1} \overline{\mathbf{b}} - \frac{1}{\sqrt{n}} \frac{1}{n} \sum_{i=1}^{n} \mathbf{D}_i^{-\frac{1}{2}} \mathbf{b}_i$ *and* $\overline{\mathbf{b}} = \frac{1}{n} \sum_{i=1}^{n} \mathbf{b}_i$.

*Proof.* By using **L**-smoothness

$$
\mathbb{E}\left[f(x^{k+1})\mid x^k\right] \overset{(8)}{\leq} f(x^k) - \gamma\left\langle\nabla f(x^k), \mathbb{E}\left[g^k\right]\right\rangle + \frac{\gamma^2}{2}\mathbb{E}\left[\|g^k\|_{\mathbf{L}}^2\right]
$$

$$
\overset{(27),(38)}{=} f(x^k) - \gamma\left\langle\nabla f(x^k), \overline{\mathbf{L}}^{-1}\nabla f(x^k) + h\right\rangle
$$

$$
+ \frac{\gamma^2}{2}\left(\|\mathbb{E}\left[g^k\right]\|_{\mathbf{L}}^2 + \mathbb{E}\left[\|g^k - \mathbb{E}\left[g^k\right]\|_{\mathbf{L}}^2\right]\right)
$$

$$
\overset{(27)}{=} f(x^k) - \gamma\left(\left\langle\nabla f(x^k), \overline{\mathbf{L}}^{-1}\nabla f(x^k)\right\rangle + \left\langle\nabla f(x^k), h\right\rangle\right)
$$

$$
+ \frac{\gamma^2}{2}\left(\left\|\overline{\mathbf{L}}^{-1}\nabla f(x^k) + h\right\|_{\overline{\mathbf{L}}}^2 + \mathbb{E}\left[\|g^k - \mathbb{E}\left[g^k\right]\|_{\mathbf{L}}^2\right]\right)
$$

$$
\overset{(37)}{=} f(x^k) - \gamma\left(\|\nabla f(x^k)\|_{\overline{\mathbf{L}}^{-1}}^2 + \left\langle\nabla f(x^k), h\right\rangle\right) + \frac{\gamma^2}{2}\mathbb{E}\left[\|g^k - \mathbb{E}\left[g^k\right]\|_{\mathbf{L}}^2\right]
$$

$$
+ \frac{\gamma^2}{2}\left(\|\nabla f(x^k)\|_{\overline{\mathbf{L}}^{-1}}^2 + 2\left\langle\nabla f(x^k), h\right\rangle + \|h\|_{\overline{\mathbf{L}}}^2\right)
$$

$$
\leq f(x^k) - \gamma\left(1 - \gamma/2\right)\|\nabla f(x^k)\|_{\overline{\mathbf{L}}^{-1}}^2 + \frac{\gamma^2}{2}\sigma^2
$$

$$
- \gamma\left(1 - \gamma\right)\left\langle\nabla f(x^k), h\right\rangle + \frac{\gamma^2}{2}\|h\|_{\overline{\mathbf{L}}}^2,
$$

where the last inequality follows from the grouping of similar terms and bounded heterogeneity

$$
\mathbb{E}\left[\|g^k - \mathbb{E}\left[g^k\right]\|_{\mathbf{L}}^2\right] = \mathbb{E}\left[\left\|g^k - \left(\overline{\mathbf{L}}^{-1}\nabla f(x^k) + h\right)\right\|_{\mathbf{L}}^2\right] \tag{50}
$$

$$
= \mathbb{E}\left[\left\|\overline{\mathbf{B}}^k x^k - \overline{\mathbf{Cb}} - \left(x^k - \frac{1}{\sqrt{n}}\widetilde{\mathbf{D}\,\mathbf{b}}\right)\right\|_{\overline{\mathbf{L}}}^2\right] \leq \sigma^2. \tag{51}
$$

Next by using a Fenchel-Young inequality (39) for $\left\langle\nabla f(x^k), -h\right\rangle$ and $1 - \gamma \geq 0$

$$
\mathbb{E}\left[f(x^{k+1})\mid x^k\right] \leq f(x^k) - \gamma\left(1 - \gamma/2\right)\|\nabla f(x^k)\|_{\overline{\mathbf{L}}^{-1}}^2 + \frac{\gamma^2}{2}\left(\|h\|_{\overline{\mathbf{L}}}^2 + \sigma^2\right)
$$

$$
+ \gamma\left(1 - \gamma\right)\left[\beta\|\nabla f(x^k)\|_{\overline{\mathbf{L}}^{-1}}^2 + 0.25\beta^{-1}\|h\|_{\overline{\mathbf{L}}}^2\right]
$$

$$
\leq f(x^k) - \gamma\left(1 - \gamma/2 - \beta\left(1 - \gamma\right)\right)\|\nabla f(x^k)\|_{\overline{\mathbf{L}}^{-1}}^2
$$

$$
+ \gamma\left\{\left(\beta^{-1}\left(1 - \gamma\right) + \frac{\gamma}{2}\right)\|h\|_{\overline{\mathbf{L}}}^2 + \frac{\gamma}{2}\sigma^2\right\}, \tag{52}
$$

where in the last inequality we grouped similar terms and used the fact that $0.25 < 1$.

Now to guarantee that $1 - \gamma/2 - \beta(1 - \gamma) \geq c > 0$ we choose the step size as

$$
0 < \gamma \leq \gamma_{c,\beta} := \frac{1 - c - \beta}{\beta + 1/2}, \tag{53}
$$

where $\gamma_{c,\beta} > 0$ for $\beta + c < 1$. This means that $\beta$ can not arbitrary grow to diminish $\beta^{-1}$.
Then after standard manipulations and unrolling the recursion

$$
\gamma c\|\nabla f(x^k)\|_{\overline{\mathbf{L}}^{-1}}^2 \leq f(x^k) - \mathbb{E}\left[f(x^{k+1})\mid x^k\right] + \gamma\left(\beta^{-1}\left(1 - \gamma\right) + \gamma/2\right)\|h\|_{\overline{\mathbf{L}}}^2 + \frac{\gamma^2}{2}\sigma^2 \tag{54}
$$

we obtain

$$
\frac{c}{K}\sum_{k=0}^{K-1}\mathbb{E}\left[\|\nabla f(x^k)\|_{\overline{\mathbf{L}}^{-1}}^2\right] \leq \frac{f(x^0) - \mathbb{E}\left[f(x^K)\right]}{\gamma K} + \left(\beta^{-1}\left(1 - \gamma\right) + \gamma/2\right)\|h\|_{\overline{\mathbf{L}}}^2 + \frac{\gamma}{2}\sigma^2. \tag{55}
$$

$\square$

### B.3.2 Homogeneous case

The main difference to the result in the previous Subsection is that gradient estimator expression (31) holds deterministically (without expectation $\mathbb{E}$). That is why $g^k = \mathbb{E}\left[g^k\right]$ and heterogeneity term $\sigma^2$ equals to 0.

We provide the full statement and proof for the homogeneous result discussed in 4.2.

**Theorem 4.** *Consider the method* (2) *with estimator* (31) *for a homogeneous quadratic problem* (12) *with positive definite matrix* $\mathbf{L}_i \equiv \mathbf{L} \succ 0$. *Then if exists* $\mathbf{D}^{-\frac{1}{2}}$ *for* $\mathbf{D} := \mathrm{Diag}(\mathbf{L})$, *scaled permutation sketch* $\mathbf{C}_i' = \sqrt{n}\, e_{\pi_i} e_{\pi_i}^\top$ *is used and the step size is chosen as*

$$0 < \gamma \le \gamma_{c,\beta} := \frac{1 - c - \beta}{\beta + 1/2}, \tag{56}$$

*where* $\gamma_{c,\beta} > 0$ *for* $\beta + c < 1$. *Then the iterates satisfy*

$$\frac{1}{K}\sum_{k=0}^{K-1}\mathbb{E}\left[\left\|\nabla f(x^k)\right\|_{\tilde{\mathbf{L}}^{-1}}^2\right] \le \frac{f(x^0) - \mathbb{E}\left[f(x^K)\right]}{c\gamma K} + \left(\frac{1-\gamma}{c\beta} + \frac{\gamma}{2c}\right)\|h\|_{\tilde{\mathbf{L}}}^2, \tag{57}$$

*where* $\tilde{\mathbf{L}} = \mathbf{D}^{-\frac{1}{2}}\mathbf{L}\mathbf{D}^{-\frac{1}{2}}, h = \tilde{\mathbf{L}}^{-1}\tilde{\mathrm{b}} - \frac{1}{\sqrt{n}}\tilde{\mathrm{b}}$ *and* $\tilde{\mathrm{b}} = \mathbf{D}^{-\frac{1}{2}}\mathrm{b}$.

*Proof.* By using $\mathbf{L}$-smoothness

$$
\begin{aligned}
\mathbb{E}\left[f(x^k - \gamma g^k)\mid x^k\right] &\overset{(8)}{\le} f(x^k) - \left\langle \nabla f(x^k), \gamma\mathbb{E}\left[g^k\right]\right\rangle + \frac{\gamma^2}{2}\mathbb{E}\left[\left\|g^k\right\|_{\tilde{\mathbf{L}}}^2\right]\\
&\le f(x^k) - \gamma\left\langle \nabla f(x^k), \tilde{\mathbf{L}}^{-1}\nabla f(x^k) + h\right\rangle + \frac{\gamma^2}{2}\left\|\tilde{\mathbf{L}}^{-1}\nabla f(x^k) + h\right\|_{\tilde{\mathbf{L}}}^2\\
&\overset{(37)}{=} f(x^k) - \gamma\left(\left\langle \nabla f(x^k), \tilde{\mathbf{L}}^{-1}\nabla f(x^k)\right\rangle + \left\langle \nabla f(x^k), h\right\rangle\right)\\
&\qquad + \frac{\gamma^2}{2}\left(\left\|\nabla f(x^k)\right\|_{\tilde{\mathbf{L}}^{-1}}^2 + 2\left\langle \nabla f(x^k), h\right\rangle + \|h\|_{\tilde{\mathbf{L}}}^2\right)\\
&= f(x^k) - \gamma(1 - \gamma/2)\left\|\nabla f(x^k)\right\|_{\tilde{\mathbf{L}}^{-1}}^2 + \frac{\gamma^2}{2}\|h\|_{\tilde{\mathbf{L}}}^2 - \gamma(1-\gamma)\left\langle \nabla f(x^k), h\right\rangle
\end{aligned}
$$

Next by using a Fenchel-Young inequality (39) for $\left\langle \nabla f(x^k), -h\right\rangle$ and $1 - \gamma \ge 0$

$$
\begin{aligned}
\mathbb{E}\left[f(x^{k+1})\mid x^k\right] &\le f(x^k) - \gamma(1-\gamma/2)\left\|\nabla f(x^k)\right\|_{\tilde{\mathbf{L}}^{-1}}^2 + \frac{\gamma^2}{2}\|h\|_{\tilde{\mathbf{L}}}^2\\
&\qquad + \gamma(1-\gamma)\left[\beta\|\nabla f(x^k)\|_{\tilde{\mathbf{L}}^{-1}}^2 + 0.25\beta^{-1}\|h\|_{\tilde{\mathbf{L}}}^2\right]\\
&= f(x^k) - \gamma(1-\gamma/2 - \beta(1-\gamma))\left\|\nabla f(x^k)\right\|_{\tilde{\mathbf{L}}^{-1}}^2\\
&\qquad + \gamma\left(\beta^{-1}(1-\gamma) + \gamma/2\right)\|h\|_{\tilde{\mathbf{L}}}^2.
\end{aligned}
$$

Now to guarantee that $1 - \gamma/2 - \beta(1-\gamma) \ge c > 0$ we choose the step size as

$$0 < \gamma \le \gamma_{c,\beta} := \frac{1 - c - \beta}{\beta + 1/2}, \tag{58}$$

where $\gamma_{c,\beta} \ge 0$ for $\beta + c < 1$.
Then after standard manipulations and unrolling the recursion

$$\gamma c\left\|\nabla f(x^k)\right\|_{\tilde{\mathbf{L}}^{-1}}^2 \le f(x^k) - \mathbb{E}\left[f(x^{k+1})\mid x^k\right] + \gamma\left(\beta^{-1}(1-\gamma) + \gamma/2\right)\|h\|_{\tilde{\mathbf{L}}}^2 \tag{59}$$

we obtain the formulated result

$$\frac{c}{K}\sum_{k=0}^{K-1}\mathbb{E}\left[\left\|\nabla f(x^k)\right\|_{\tilde{\mathbf{L}}^{-1}}^2\right] \le \frac{f(x^0) - \mathbb{E}\left[f(x^K)\right]}{\gamma K} + \left(\beta^{-1}(1-\gamma) + \gamma/2\right)\|h\|_{\tilde{\mathbf{L}}}^2. \tag{60}$$

$\square$

**Remark 2.** *1) The first term in the convergence upper bound* (57) *is minimized by maximizing product $c \cdot \gamma$, which motivates to choose $c > 0$ and $\gamma \leq 1$ as big as possible. Although due to the constraint on the step size (and $\beta > 0$)*

$$0 < \gamma \leq \gamma_{c,\beta} := \frac{1 - c - \beta}{\beta + 1/2}, \tag{61}$$

*constant $c \in (0, 1)$. So, by maximizing $c$ the value $\gamma_{c,\beta}$ becomes smaller, thus there is a trade-off.*

*2) The second term or the neighborhood size (multiplier in front of $\|h\|_{\tilde{\mathbf{L}}}^2$)*

$$\Psi(\beta, \gamma) := \frac{\beta^{-1} (1 - \gamma) + \gamma/2}{c} = \frac{\beta^{-1} (1 - \gamma) + \gamma/2}{1 - \gamma/2 - \beta(1 - \gamma)} \tag{62}$$

*can be numerically minimized (e.g. by using WolframAlpha) with constraints $\gamma \in (0, 1]$ and $\beta > 0$. The solution of such optimization problem is $\gamma^\star \approx 1$ and $\beta^\star \approx \xi \in \{3.992, 2.606, 2.613\}$. In fact, $\Psi(\beta^\star, \gamma^\star) \approx 0.5$.*

**Functional gap convergence.** Note that for quadratic optimization problem (12)

$$\left\| \nabla f(x^k) \right\|_{\tilde{\mathbf{L}}^{-1}}^2 = \left\langle \tilde{\mathbf{L}} x^k - \tilde{b}, \tilde{\mathbf{L}}^{-1} \left( \tilde{\mathbf{L}} x^k - \tilde{b} \right) \right\rangle = 2 \left( f(x^k) - f(x^\star) \right). \tag{63}$$

Then by rearranging and subtracting $f^\star := f(x^\star)$ from both sides of inequality (59) we obtain

$$
\begin{aligned}
\mathbb{E}\left[ f(x^{k+1}) \mid x^k \right] - f^\star &\leq f(x^k) - f^\star - \gamma c \left\| \nabla f(x^k) \right\|_{\tilde{\mathbf{L}}^{-1}}^2 + \gamma \left( \beta^{-1} (1 - \gamma) + \gamma/2 \right) \|h\|_{\tilde{\mathbf{L}}}^2 \\
&\stackrel{(63)}{=} \left( f(x^k) - f^\star \right) - \gamma c \cdot 2 \left( f(x^k) - f^\star \right) + \gamma \left( \beta^{-1} (1 - \gamma) + \gamma/2 \right) \|h\|_{\tilde{\mathbf{L}}}^2 \\
&= (1 - 2\gamma c) \left( f(x^k) - f^\star \right) + \gamma \left( \beta^{-1} (1 - \gamma) + \gamma/2 \right) \|h\|_{\tilde{\mathbf{L}}}^2.
\end{aligned}
$$

After unrolling the recursion

$$
\begin{aligned}
\mathbb{E}\left[ f(x^{k+1}) \mid x^k \right] - f^\star &\leq (1 - 2\gamma c)^k \left( f(x^0) - f^\star \right) + \gamma \left( \beta^{-1} (1 - \gamma) + \gamma/2 \right) \|h\|_{\tilde{\mathbf{L}}}^2 \sum_{i=0}^{k} (1 - 2\gamma c)^i \\
&\leq (1 - 2\gamma c)^k \left( f(x^0) - f^\star \right) + \frac{1}{2c} \left( \beta^{-1} (1 - \gamma) + \gamma/2 \right) \|h\|_{\tilde{\mathbf{L}}}^2.
\end{aligned}
$$

This result is formalized in the following Theorem.

**Theorem 5.** *Consider the method* (2) *with estimator* (31) *for a homogeneous quadratic problem* (12) *with positive definite matrix $\mathbf{L}_i \equiv \mathbf{L} \succ 0$. Then if exists $\mathbf{D}^{-\frac{1}{2}}$ for $\mathbf{D} := \mathrm{Diag}(\mathbf{L})$, scaled permutation sketch $\mathbf{C}_i' = \sqrt{n} e_{\pi_i} e_{\pi_i}^\top$ is used and the step size is chosen as*

$$0 < \gamma \leq \gamma_{c,\beta} := \frac{1 - c - \beta}{\beta + 1/2}, \tag{64}$$

*where $\gamma_{c,\beta} > 0$ for $\beta + c < 1$. Then the iterates satisfy*

$$\mathbb{E}\left[ f(x^k) \right] - f^\star \leq (1 - 2\gamma c)^k \left( f(x^0) - f^\star \right) + \frac{1}{2c} \left( \beta^{-1} (1 - \gamma) + \gamma/2 \right) \|h\|_{\tilde{\mathbf{L}}}^2, \tag{65}$$

*where $h = \tilde{\mathbf{L}}^{-1} \tilde{b} - \frac{1}{\sqrt{n}} \tilde{b}$ and $\tilde{\mathbf{L}} = \mathbf{D}^{-\frac{1}{2}} \mathbf{L} \mathbf{D}^{-\frac{1}{2}}, \tilde{b} = \mathbf{D}^{-\frac{1}{2}} b$.*

This result shows that for a proper choice of the step size $\gamma = 1$ and constant $c = 1/2$, the functional gap can converge in basically 1 iteration to the neighborhood of size

$$\|h\|_{\tilde{\mathbf{L}}}^2 = \left\langle \tilde{\mathbf{L}} \left( \tilde{\mathbf{L}}^{-1} \tilde{b} - \frac{1}{\sqrt{n}} \tilde{b} \right), \tilde{\mathbf{L}}^{-1} \tilde{b} - \frac{1}{\sqrt{n}} \tilde{b} \right\rangle,$$

which equals to zero if $\tilde{\mathbf{L}}^{-1} \tilde{b} = \frac{1}{\sqrt{n}} \tilde{b}$. This condition is the same as the one we obtained at the end of Subsection 4.2 with asymptotic analysis of the iterates in the homogeneous case.

**Discussion of the trace.** Consider a positive definite $\mathbf{L} \succ 0$ such that $\exists \mathbf{D}^{-\frac{1}{2}}$. Thus $\tilde{\mathbf{L}} = \mathbf{D}^{-\frac{1}{2}} \mathbf{L} \mathbf{D}^{-\frac{1}{2}}$ has only ones on the diagonal and $\operatorname{tr}(\tilde{\mathbf{L}}) = n$. Then

$$n \cdot \operatorname{tr}(\tilde{\mathbf{L}}^{-1}) = \operatorname{tr}(\tilde{\mathbf{L}})\operatorname{tr}(\tilde{\mathbf{L}}^{-1}) = (\lambda_1 + \cdots + \lambda_n)\left(\frac{1}{\lambda_1} + \cdots + \frac{1}{\lambda_n}\right) \geq n^2,$$

where the last inequality is due to the relation between harmonic and arithmetic means. Therefore $\operatorname{tr}(\tilde{\mathbf{L}}^{-1}) = \lambda_1^{-1} + \cdots + \lambda_n^{-1} \geq n$ and sum of $\tilde{\mathbf{L}}^{-1}$ eigenvalues has to be greater than $n$.

## B.4 Generalization to $n \neq d$ case.

Our results can be generalized in a similar way as in [40].

**1)** $d = qn$, for integer $q \geq 1$. Let $\pi = (\pi_1, \ldots, \pi_d)$ be a random permutation of $\{1, \ldots, d\}$. Then for each $i \in \{1, \ldots, n\}$ define

$$\mathbf{C}_i' := \sqrt{n} \cdot \sum_{j=q(i-1)+1}^{qi} e_{\pi_j} e_{\pi_j}^\top. \tag{66}$$

Matrix $\mathbb{E}\left[\overline{\mathbf{B}}^k\right]$ for the homogeneous preconditioned case can be computed the following way

$$
\begin{aligned}
\mathbb{E}\left[\overline{\mathbf{B}}^k\right] &= \mathbb{E}\left[\frac{1}{n}\sum_{i=1}^n \mathbf{C}_i' \tilde{\mathbf{L}} \mathbf{C}_i'\right] \\
&= \frac{1}{n}\sum_{i=1}^n \mathbb{E}\left[\sum_{j=q(i-1)+1}^{qi} n e_{\pi_j} e_{\pi_j}^\top \tilde{\mathbf{L}} e_{\pi_j} e_{\pi_j}^\top\right] \\
&= \sum_{i=1}^n \sum_{j=q(i-1)+1}^{qi} \mathbb{E}\left[e_{\pi_j} e_{\pi_j}^\top \tilde{\mathbf{L}} e_{\pi_j} e_{\pi_j}^\top\right] \\
&= \sum_{i=1}^n \sum_{j=q(i-1)+1}^{qi} \frac{1}{d}\sum_{l=1}^d e_l e_l^\top \tilde{\mathbf{L}} e_l e_l^\top \\
&= \sum_{i=1}^n \sum_{j=q(i-1)+1}^{qi} \frac{1}{d}\operatorname{Diag}(\tilde{\mathbf{L}}) \\
&= n\frac{q}{d}\operatorname{Diag}(\tilde{\mathbf{L}}) \\
&= \operatorname{Diag}(\tilde{\mathbf{L}}) \\
&= \mathbf{I}.
\end{aligned}
$$

As for the linear term

$$
\begin{aligned}
\mathbb{E}\left[\mathbf{C}' \mathbf{b}\right] &= \mathbb{E}\left[\frac{1}{n}\sum_{i=1}^n \mathbf{C}_i' \tilde{\mathbf{b}}\right] = \frac{1}{n}\sum_{i=1}^n \mathbb{E}\left[\sum_{j=q(i-1)+1}^{qi} \sqrt{n} e_{\pi_j} e_{\pi_j}^\top \tilde{\mathbf{b}}\right] \\
&= \frac{1}{\sqrt{n}}\sum_{i=1}^n \sum_{j=q(i-1)+1}^{qi} \frac{1}{d}\mathbf{I}\tilde{\mathbf{b}} = \frac{\sqrt{n}q}{d}\mathbf{I}\tilde{\mathbf{b}} = \frac{1}{\sqrt{n}}\tilde{\mathbf{b}}.
\end{aligned}
$$

**2)** $n = qd$, for integer $q \geq 1$. Define the multiset $S := \{1, \ldots, 1, 2, \ldots, 2, \ldots, d, \ldots, d\}$, where each number occurs precisely $q$ times. Let $\pi = (\pi_1, \ldots, \pi_n)$ be a random permutation of $S$. Then for each $i \in \{1, \ldots, n\}$ define

$$\mathbf{C}_i' := \sqrt{d} \cdot e_{\pi_i} e_{\pi_i}^\top. \tag{67}$$

$$\mathbb{E}\left[\overline{\mathbf{B}}^k\right] = \mathbb{E}\left[\frac{1}{n}\sum_{i=1}^{n}\mathbf{C}_i'\tilde{\mathbf{L}}\,\mathbf{C}_i'\right] = \frac{1}{n}\sum_{i=1}^{n}\mathbb{E}\left[d e_{\pi_i}e_{\pi_i}^{\top}\tilde{\mathbf{L}}\,e_{\pi_i}e_{\pi_i}^{\top}\right]$$

$$= \frac{1}{n}\sum_{i=1}^{n}\frac{1}{d}\sum_{j=1}^{d}d e_j e_j^{\top}\tilde{\mathbf{L}}\,e_j e_j^{\top} = \frac{1}{n}\sum_{i=1}^{n}\mathrm{Diag}(\tilde{\mathbf{L}}) = \mathbf{I}.$$

The linear term

$$\mathbb{E}\left[\mathbf{C}'\,\mathrm{b}\right] = \mathbb{E}\left[\frac{1}{n}\sum_{i=1}^{n}\mathbf{C}_i'\,\tilde{\mathrm{b}}\right] = \frac{1}{n}\sum_{i=1}^{n}\mathbb{E}\left[\sqrt{d}e_{\pi_i}e_{\pi_i}^{\top}\,\tilde{\mathrm{b}}\right] = \frac{\sqrt{d}}{n}\sum_{i=1}^{n}\frac{1}{d}\mathbf{I}\,\tilde{\mathrm{b}} = \frac{1}{\sqrt{d}}\,\tilde{\mathrm{b}}\,.$$

To sum up both cases, in a homogeneous preconditioned setting $\mathbb{E}\left[\overline{\mathbf{B}}^k\right] = \mathbf{I}$ and

$$\mathbb{E}\left[\mathbf{C}'\,\mathrm{b}\right] = \mathbb{E}\left[\frac{1}{n}\sum_{i=1}^{n}\mathbf{C}_i'\,\mathrm{b}\right] = \tilde{\mathrm{b}}\,/\sqrt{\min(n,d)}.$$

Similar modifications and calculations can be done for heterogeneous scenario. The case when $n$ does not divide $d$ and vice versa is generalized using constructions from [40].

# C  Comparison to previous related works

**Overview of theory provided in the original IST work [45].**  They consider the following method

$$x^{k+1} = \mathcal{C}(x^k) - \gamma\nabla f_{i_k}(\mathcal{C}(x^k)), \tag{68}$$

where $[\mathcal{C}(x)]_i = x_i \cdot \mathcal{B}e(p)$[4] is Bernoulli sparsifier and $i_k$ is sampled uniformly at random from $[n]$.

Analysis in [45] relies on the assumptions

1. $L_i$-smoothness of individual losses $f_i$;

2. $Q$-Lipschitz continuity of $f$: $|f(x) - f(y)| \le Q\|x - y\|$;

3. Error bound (or PŁ-condition): $\|\nabla f(x)\| \ge \mu\|x^\star - x\|$, where $x^\star$ is the global optimum;

4. Stochastic gradient variance: $\mathbb{E}\left[\|\nabla f_{i_k}(x)\|^2\right] \le M + M_f\|\nabla f(x)\|^2$;

5. $\mathbb{E}\left[\nabla f_{i_k}(\mathcal{C}(x^k))\,|\,x^k\right] = \nabla f(x^k) + \varepsilon,\quad \|\varepsilon\| \le B.$

Convergence result [45, Theorem 1] for step size $\gamma = {1}/{(2L_{\max})}$:

$$\min_{k\in\{1,\dots,K\}}\mathbb{E}\left[\|\nabla f(x^k)\|^2\right] \le \frac{f(x^0) - f(x^\star)}{\alpha(K+1)} + \frac{1}{\alpha}\cdot\left(\frac{BQ}{2L_{\max}} + \frac{5L_{\max}\omega}{2}\|x^\star\|^2 + \frac{M}{4L_{\max}}\right), \tag{69}$$

where $\alpha := \frac{1}{2L_{\max}}\left(1 - \frac{M_f}{2}\right) - \frac{5\omega L_{\max}}{2\mu^2}$, $\omega := \frac{1}{p} - 1 < \frac{\mu^2}{10L_{\max}^2}$, and $L_{\max} := \max_i L_i$.

If Lipschitzness and Assumption 5 is replaced with *norm condition*:

$$\|\mathbb{E}\left[\nabla f_{i_k}(\mathcal{C}(x^k))\,|\,x^k\right] - \nabla f(x^k)\| \le \theta\|\nabla f(x^k)\| \tag{70}$$

they obtain the following (for step size $\gamma = {1}/{2L_{\max}}$)

$$\min_{k\in\{1,\dots,K\}}\mathbb{E}\left[\|\nabla f(x^k)\|^2\right] \le \frac{f(x^0) - f(x^\star)}{\alpha(K+1)} + \frac{1}{\alpha}\cdot\left(\frac{5L_{\max}\omega}{2}\|x^\star\|^2 + \frac{M}{4L_{\max}}\right), \tag{71}$$

where $\alpha = \frac{1}{2L_{\max}}\left(\frac{1}{2} - \theta - \frac{M_f}{2}\right) - \frac{5\omega L_{\max}}{2\mu^2}$ and $\omega = \frac{1}{p} - 1 < \frac{\mu^2}{5L_{\max}^2\left(\frac{1}{2}-\theta-\frac{M_f}{2}\right)}$.

---

[4] $\mathcal{B}_p(x) := \begin{cases} x/p & \text{with probability } p \\ 0 & \text{with probability } 1 - p \end{cases}$

**Remark 3.** *The method* (68) *does not incorporate gradient sparsification, which can create a significant disparity between theory and practice. This is because the gradient computed at the compressed model, denoted as $\nabla f(\mathcal{C}(x))$, is not guaranteed to be sparse and representative of submodel computations. Such modification of the method also significantly simplifies theoretical analysis, as using a single sketch (instead of* **CLC***) allows for an unbiased gradient estimator.*

*Through our analysis of the IST gradient estimator in Equation* (31)*, we discover that conditions, such as Assumption 5, are not satisfied even in the homogeneous setting for a simple quadratic problem. Furthermore, it is evident that such conditions are also not met for logistic loss. At the same time, in generally, it is expected that insightful theory for general (non-)convex functions should yield appropriate results for quadratic problems. Additionally, it remains unclear whether the norm condition* (70) *is satisfied in practical scenarios, as even for quadratic problems, the situation is not straightforward, as we show in the expression for $\sigma^2$ in* (50)*.*

**Masked training [33].** The authors consider the following "Partial SGD" method

$$
\begin{aligned}
\hat{x}^k &= x^k + \delta x^k = x^k - (1-p) \odot x^k \\
x^{k+1} &= x^k - \gamma p \odot \nabla f(\hat{x}^k, \xi^k),
\end{aligned}
\tag{72}
$$

where $\nabla f(x, \xi)$ is an unbiased stochastic gradient estimator of an $L$-smooth loss function $f$, $\odot$ is an element-wise product, and $p$ is a binary sparsification mask.

They make the following "bounded perturbation" assumption

$$
\max_k \frac{\|\delta x^k\|}{\max\{\|p^k \odot \nabla f(x^k)\|, \|p^k \odot \nabla f(\hat{x}^k)\|\}} \le \frac{1}{2L}.
\tag{73}
$$

This assumption may not hold for a simple convex case. Consider a quadratic function $f(x) = \frac{1}{2} x^\top A x$, for

$$
A = \begin{pmatrix} a & 0 \\ 0 & c \end{pmatrix}, \qquad x^0 = \begin{pmatrix} x_1 \\ x_2 \end{pmatrix}, \qquad p^0 = \begin{pmatrix} 0 \\ 1 \end{pmatrix}.
\tag{74}
$$

Then condition (73) (at iteration $k = 0$) will be equivalent to

$$
\frac{x_1}{cx_2} \le \frac{1}{2a} \Leftrightarrow 2 \le \frac{2a}{c} \le \frac{x_2}{x_1},
$$

which clearly does not hold for an arbitrary initialization $x^0$.

In addition, convergence bound in [33, Theorem 1] suggests choosing the step size as $\gamma_0 \alpha^k$, where

$$
\alpha^k = \min\left\{1, \frac{\langle p^k \odot \nabla f(x^k), p^k \odot \nabla f(\hat{x}^k)\rangle}{\|p^k \odot \nabla f(\hat{x}^k)\|^2}\right\}
\tag{75}
$$

is not guaranteed to be positive to the inner product $\langle p^k \odot \nabla f(x^k), p^k \odot \nabla f(\hat{x}^k)\rangle$, which may lead to non-convergence of the method.

**Optimization with access to auxiliary information framework [7]** suggests modeling training with compressed models via performing gradient steps with respect to function $h(x) :=  \mathbb{E}_{\mathcal{M}}[f(1_{\mathcal{M}} \odot x)]$. This function allows access to sparse/low-rank version of the original model $f(x)$. They impose the following bounded hessian dissimilarity assumption on $h$ and $f$

$$
\left\|\nabla^2 f(x) - \mathbb{E}_{\mathcal{M}}\left[\mathbf{D}_{\mathcal{M}} \nabla^2 f(1_{\mathcal{M}} \odot x) \mathbf{D}_{\mathcal{M}}\right]\right\|_2 \le \delta,
\tag{76}
$$

where $1_{\mathcal{M}}$ and $\mathbf{D}_{\mathcal{M}} = \mathrm{Diag}(1_{\mathcal{M}})$ refer to binary vector and matrix sparsification masks.

This approach relies on variance-reduction and requires gradient computations on the full model $x$. That is why it is not suitable for our problem setting.

 # D   Experiments

To empirically validate our theoretical framework and its implications, we focus on carefully controlled settings that satisfy the assumptions of our work. Specifically, we consider a quadratic problem defined in (12). For reminder, the local loss function is defined as

$$f_i(x) = x^\top \mathbf{L}_i x - x^\top \mathrm{b}_i,$$

where $\mathbf{L}_i = \mathbf{B}_i^\top \mathbf{B}_i$. Entries of the matrices $\mathbf{B}_i \in \mathbb{R}^{d \times d}$, vectors $\mathrm{b}_i \in \mathbb{R}^d$, and initialization $x^0 \in \mathbb{R}^d$ are generated from a standard Gaussian distribution $\mathcal{N}(0, 1)$.

In Figure 1a, we present the performance of the simplified Independent Subnetwork Training (IST) algorithm (update (2) with estimator (26)) for a heterogeneous problem. We fix the dimension $d$ to 1000 and the number of computing nodes $n$ to 10. We evaluate the logarithm of a relative functional error $\log\left(f(x^k) - f(x^\star)/f(x^0) - f(x^\star)\right)$, while the horizontal axis denotes the number of communication rounds required to achieve a certain error tolerance. According to our theory (65), the method converges to a neighborhood of the solution, which depends on the chosen step size. Specifically, a larger step size allows for faster convergence but results in a larger neighborhood.

In Figure 1b, we demonstrate the convergence of the iterates $x^k$ for a homogeneous problem with $d = n = 50$. The results are in close agreement with our theoretical predictions for the estimator (31). We observe that the distance to the method's expected fixed point $x^\infty = \tilde{\mathrm{b}}/\sqrt{n}$ decreases linearly for different step size values. This confirms that IST may converge not the optimal solution $x^\star = \tilde{\mathbf{L}}^{-1}\tilde{\mathrm{b}}$ of the original problem (12) in general (no interpolation) case.

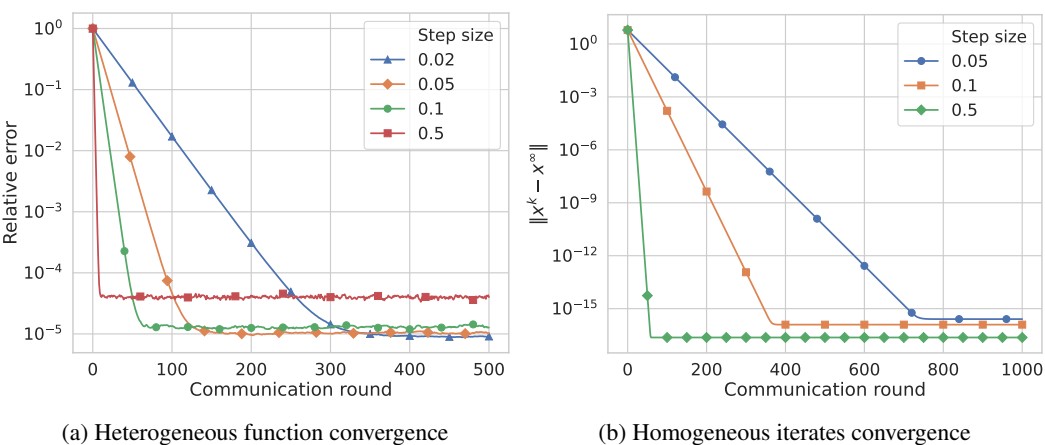

(a) Heterogeneous function convergence     (b) Homogeneous iterates convergence

Figure 1: Different step size values

Simulations were performed on a machine with $24\,\mathrm{Intel(R)}\,\mathrm{Xeon(R)}\,\mathrm{Gold}\,6246\,\mathrm{CPU}\,@\,3.30\,\mathrm{GHz}$.