# OpenReview forum: "Towards a Better Theoretical Understanding of Independent Subnetwork Training"
_NeurIPS.cc/2023/Conference — Submitted to NeurIPS 2023_

### Official Review · Reviewer_ijy7 · 2023-07-01

**Soundness:** 3 good
**Presentation:** 3 good
**Contribution:** 3 good
**Rating:** 6
**Confidence:** 2

**Summary:**

The authors provided a theoretical analysis for independent subnet training and provided a convergence analysis in the case where communication compression is present.

The authors discussed the scenario when bias is not present and provided two analyses in the homogeneous and heterogeneous case, respectively before extending their theorems to the case with bias.

**Strengths:**

1. Sections 3 and 4 are well written and clear, which break the scenarios down in an intuitive way and presented the theorems and outlines clearly.

**Weaknesses:**

1. Introduction could be more straightforward and can dive straight into the main technical contributions of the work. It was not clear why the problem is well motivated and what are the main technical hurdles until reading section 2 and onwards. A clearer presentation in the intro can make the paper much more readable and well-motivated.

**Questions:**

1. The authors mentioned that prior work on the convergence of IST focus "on overparameterized single hidden layer neural networks with ReLU activations". It is not entirely clear to me why the authors considered quadratic form and what is the tradeoff between their work and prior work in this regard. A more thorough explanation would be appreciated.

**Limitations:**

The work is theoretical and does not seem to have any potential negative societal impact.

---

> ### Author Rebuttal · Authors · 2023-08-07
>
> Dear Reviewer LM8J,
>
> Thanks for the time and effort devoted to our paper. We appreciate the positive evaluation of our work.
>
> # Responses to questions
> > The authors mentioned that prior work on the convergence of IST focus "on overparameterized single hidden layer neural networks with ReLU activations". It is not entirely clear to me why the authors considered quadratic form and what is the tradeoff between their work and prior work in this regard. A more thorough explanation would be appreciated.
>
> We have tried to clarify the key differences between our approaches in general [response](https://openreview.net/forum?id=ldulVsMDDk&noteId=aaNJQt5Ksk) to all reviewers.
>
> ## Comments on Weaknesses
> > Introduction could be more straightforward and can dive straight into the main technical contributions of the work. It was not clear why the problem is well motivated and what are the main technical hurdles until reading section 2 and onwards. A clearer presentation in the intro can make the paper much more readable and well-motivated.
>
> Thank you for the suggestion. We will try to modify the Introduction according to your proposal.
>
> Best regards,
>
> Authors

---

### Official Review · Reviewer_LM8J · 2023-07-02

**Soundness:** 3 good
**Presentation:** 3 good
**Contribution:** 3 good
**Rating:** 7
**Confidence:** 4

**Summary:**

The paper provides a theoretical analysis of the convergence properties of Independent Subnetwork Training (IST), for distributed Stochastic Gradient Descent (SGD) optimization with a quadratic loss function. The analysis considers both the cases of homogeneous and heterogeneous distributed scenarios, without restrictive assumptions on the gradient estimator.

The work characterizes situations where IST converges very efficiently, and cases where it does not converge to the optimal solution but to an irreducible neighbourhood. Experimental results that validate the theory are provided in the Appendix.

**Strengths:**

The paper provides a solid analytical treatment of the important problem of distributed optimization with reduced communication overhead by means of Independent Subnetwork Training (IST).

Compared to previous work, the analysis of the paper does not rely on the restrictive assumption of a bounded stochastic gradient norm.

The paper is well written – the exposition is clear, and the material is well structured and well presented.

**Weaknesses:**

The work considers distributed Stochastic Gradient Descent (SGD) training with a quadratic loss. As mentioned by the authors in Section 3, a simple quadratic loss function has been used in other work to analyze properties of neural networks. While this loss function can still provide interesting theoretical insights, it would be valuable to extend the analysis and the experimental results to more generally used loss functions.

Minor comments:

-- Line 17: "drives from" may be changed to "derives from".

-- Equation after line 217: it seems that the first part of the equation
"$\mathbb{E}[g^k] = \bar{\mathbf{L}}^{-1} \bar{\mathbf{L}} x^k \pm \bar{\mathbf{L}}^{-1} \bar{b} - \frac{1}{\sqrt{n}} \tilde{\mathbf{D} b} = $ ..."
should be rewritten as
"$\mathbb{E}[g^k] = \bar{\mathbf{L}}^{-1} \bar{\mathbf{L}} x^k - \frac{1}{\sqrt{n}} \tilde{\mathbf{D} b} = $ ...".

-- Equation after line 221: it seems that the first part of the equation
"$\mathbb{E}[x^{k+1}] = x^k - \gamma \mathbb{E}[g^k] = $ ..."
should be
"$\mathbb{E}[x^{k+1}] = \mathbb{E}[x^k] - \gamma \mathbb{E}[g^k] = $ ...".



**Questions:**

It would be valuable if the authors could consider a discussion or possible additional experiments to extend some of the results and insights presented in the paper to the case of distributed optimization with more generally used loss functions.

**Limitations:**

The work relates to distributed training of large-scale models, which generally correspond to significant power consumption and CO2 emissions. However, the IST method studied in the paper aims at allowing distributed training with reduced communication overhead, corresponding to potentially reduced power consumption.

---

> ### Author Rebuttal · Authors · 2023-08-07
>
> Dear Reviewer LM8J,
>
> Thanks for the time and effort devoted to our paper. We greatly value careful reading of the material and the positive evaluation of our work.
>
> # Responses to questions
> > It would be valuable if the authors could consider a discussion or possible additional experiments to extend some of the results and insights presented in the paper to the case of distributed optimization with more generally used loss functions.
>
> Thank you for this suggestion. Let us comment on the matter of generalizing our results to non-quadratic settings.
> First, we note that there were already attempts [45, 48] to perform analysis of similar classes of methods for a different class of loss functions. There is a discussion of their results in Section 4.3 and Appendix C in more detail. We find their convergence bounds unsatisfying from a theoretical viewpoint due to too restrictive additional assumptions (e.g., on bounded gradient norm or sparsification parameter), which lead to vacuous bounds in certain cases. That is why we decided to take a step back and start with a quadratic problem setting, which allowed us to perform a meaningful theoretical analysis and reveal the advantages and limitations of IST. Generalizing our results to a non-quadratic setting is currently open, and it is not entirely clear how difficult the problem is. When we tried to solve this issue ourselves, it was found that we are not aware of a theoretical framework that allows to perform analysis for a general class of $L$-smooth functions due to challenges (e.g. biased gradient estimator) mentioned in Section 2.1 and before 4.3.
>
> ## Clarifications on “minor comments”
>
> > -- Equation after line 221:
>
> Thank you for this comment. Let us clear this up. On line 221, we mention “proper conditioning”, which means applying the following conditional expectation $\mathbb{E}\left[\cdot | x^k \right]$. That being said, a more formal form of equation (28) on line 221 is
> $$
> \mathbb{E}\left[x^{k+1} | x^k\right] = \mathbb{E}\left[x^k\right]-\gamma \mathbb{E}\left[g^k | x^k\right].
> $$
> So there is no randomness in $x^k$ on Line 221.
>
> > -- Equation after line 217:
>
> Thank you for the suggestion. Let us clarify what is meant there. The transformation that is done there is based on the idea of “smart” zero: adding and subtracting the same term
> $$
> \mathbb{E}\left[g^k | x^k\right] = \overline{\mathbf{L}}^{-1} \overline{\mathbf{L}} x^k - \frac{1}{\sqrt{n}} \widetilde{\mathbf{D} \mathrm{b}} = \overline{\mathbf{L}}^{-1} \overline{\mathbf{L}} x^k - \overline{\mathbf{L}}^{-1} \overline{\mathrm{b}} -\frac{1}{\sqrt{n}} \widetilde{\mathbf{D} \mathrm{b}} + \overline{\mathbf{L}}^{-1} \overline{\mathrm{b}} = \overline{\mathbf{L}}^{-1} \nabla f\left(x^k\right) - \frac{1}{\sqrt{n}} \widetilde{\mathbf{D b}} + \overline{\mathbf{L}}^{-1} \overline{\mathrm{b}}.
> $$
>
> Best regards,
>
> Authors

---

> > ### Comment · Reviewer_LM8J · 2023-08-14
> >
> > I would like to thank the authors for addressing the points raised in my review and for proposing to clarify the exposition. I confirm that I am satisfied with their answers provided by the authors in the rebuttal.

---

> > > ### Author Response · Authors · 2023-08-14
> > > **Thanks!**
> > >
> > > Thanks!

---

### Official Review · Reviewer_wecA · 2023-07-04

**Soundness:** 2 fair
**Presentation:** 2 fair
**Contribution:** 2 fair
**Rating:** 4
**Confidence:** 3

**Summary:**

Independent Subnetwork Training (IST) is a technique that divides the neural network into smaller independent subnetworks, trains them in a distributed parallel way, and aggregates the results of each independent subnetwork to update the weights of the whole model.
This paper aims to analyze the behavior of IST theoretically. Specifically, it considers a quadratic model trained by IST. It conducts convergence analysis under both homogeneous and heterogeneous scenarios and shows that IST can only converge to an irreducible neighborhood of optimal solution.


**Strengths:**

This is probably the first work providing a thorough theoretical analysis of IST.

**Weaknesses:**

1. This paper should include a more comprehensive motivation for the theoretical study of IST. This could involve discussing the potential limitations of current IST architectures and how a theoretical analysis can guide future modifications to improve their performance. By doing so, reviewers will have a clearer understanding of the significance of the paper's findings and how they can be applied in practice.
2. The main body of this paper does not have any experimental results. The authors should include some key experiments to validate their theoretical analysis.
3. The authors should consider expanding the scope of their experiments beyond quadratic models to include other types of models that are commonly used in SOTA IST papers, e.g., ResNet, and Graph Convolutional Networks, as listed in this paper's reference. This would allow reviewers to better understand the generality of the paper's findings and how they can be applied to real-world applications.


**Questions:**

1. How do the findings of this work be of any help to the future design of IST architecture? In addition, the authors should include more experiments on SOTA IST applications, e.g., ResNet, and Graph Convolutional Networks, to indicate the generality and significance of this paper's findings.
2. The authors should clarify the assumptions made in the permutation example in Section 3.1. Specifically, they should explain the case that n=d^2 and the use of Perm-1 sketch, while according to definition 2, perm-q refers to d=qn, which leads to n=d=1 and it’s a naïve configuration. By clarifying the assumptions made in this section, the reader can better understand the example and its implications for IST.


**Limitations:**

Future works are discussed in the conclusion section.

---

> ### Author Rebuttal · Authors · 2023-08-07
>
> Dear Reviewer wecA,
>
> Thanks for the time and effort devoted to our paper.
>
> # Responses to questions
> > How do the findings of this work be of any help to the future design of IST architecture? In addition, the authors should include more experiments on SOTA IST applications, e.g., ResNet, and Graph Convolutional Networks, to indicate the generality and significance of this paper's findings.
>
> 1. Our main goal is to formalize the IST problem setting mathematically and provide meaningful convergence theory. We are making the first step in rigorously understanding this family of algorithms that combine model and data parallelism, which has almost never been studied.
> This work is not trying to suggest a new neural network architecture for IST. We focus on analyzing the optimization (and not generalization) aspects. Our insights reveal the advantages and limitations of IST:
>
> - In the interpolation case (often considered to apply to modern Deep Learning models), IST can perform very efficiently in both homogeneous and heterogeneous scenarios.
>
> - In the more general case, we show that naïve/vanilla IST’s convergence heavily depends on the heterogeneity level, which may require decreasing the step size later throughout the optimization. While in a data center (local cluster) setting, this can be fixed by access to shared data, in federated learning (very heterogeneous), a different method may be needed.
>
> We believe that to provide guidance for the future design of IST, first, it is important to understand its basic and fundamental properties. There are plenty of works mentioned in the Introduction part of our paper, which explore the experimental aspects of IST. That is why the motivation of this work is a lack of comprehension regarding the theoretical properties of methods that perform training with compressed models.
>
> Additionally, we would like to highlight that the main contribution of our work is from the theory side, and that is why in our experiments, we focus on well-controlled settings which satisfy the assumptions in our paper to provide evidence that our theory translates into observable predictions. These are well-designed experiments that do support our theory and main claims. Since our results guarantee that the methods work, we do not need to test them extensively on large or complicated datasets and models to show that they do (which is clearly necessary for heuristics not supported by any theory). Our goal was not to claim practical SOTA on some benchmarks - that's not what our paper is about.
>
> > The authors should clarify the assumptions made in the permutation example in Section 3.1. Specifically, they should explain the case that n=d^2 and the use of Perm-1 sketch, while according to definition 2, perm-q refers to d=qn, which leads to n=d=1 and it’s a naïve configuration. By clarifying the assumptions made in this section, the reader can better understand the example and its implications for IST.
>
> 2. Thank you for pointing this out. Please note that it is not the power 2 (for $n=d^2$) but a footnote link. So $n=d$ is meant there; thus, there is no problem leading to “naïve configuration” $n=d=1$. We have already fixed this issue in the revision of the paper.
>
> ## Comments on Weaknesses
>
> > The main body of this paper does not have any experimental results. The authors should include some key experiments to validate their theoretical analysis.
>
> We would like to highlight that the title of our work is _“Towards a Better **Theoretical** Understanding of IST”_. That is why, in the main part of the paper, we focus on contributions coming from theoretical analysis. Another reason is the limited space allowed by the conference submission format. If the reviewer believes that moving the experimental results to the main part will strengthen the points we are making, we can do this in the camera-ready version.
>
> Regards,
>
> Authors

---

> > ### Comment · Reviewer_wecA · 2023-08-16
> >
> > Thanks for the clarification. The contributions and limitations of this work are clear now. The reviewer does not have further questions.

---

> > > ### Author Response · Authors · 2023-08-16
> > > **No further questions**
> > >
> > > Thanks for confirming.
> > >
> > > If you are satisfied with the replies, please consider raising your score accordingly.
> > >
> > > Thanks!
> > >
> > > authors

---

> > > > ### Comment · Reviewer_wecA · 2023-08-18
> > > >
> > > > Thanks for asking. The reviewer has decided to keep the rating.

---

### Official Review · Reviewer_87cd · 2023-07-26

**Soundness:** 3 good
**Presentation:** 3 good
**Contribution:** 2 fair
**Rating:** 5
**Confidence:** 3

**Summary:**

This submission presents a theoretical analysis for the independent subnetwork training (IST) algorithm for training models under both data and model parallel settings. The convergence guarantees are analyzed for quadratic loss functions, when using permutation sketches as the model compressors.

**Strengths:**

- Optimization using data and model parallelization is an important practical problem, for which more theoretical insights are welcome
- The submission presents a fairly thorough analysis of the convergence guarantees for IST for the quadratic loss function analyzed
- The authors highlight some limitations of IST, which would be useful to be aware of (e.g. the fact that in the general case of the quadratic function the algorithm does not converge to the solution)
- The submission is overall well written, and the authors are careful to introduce the setup and all the assumptions used in their analysis


**Weaknesses:**

- Overall the analysis presented in this submission is very limited, as it only deals with a specific type of loss function, which is not a practical instance where model parallelization would be useful
- While the authors argue that the quadratic model has been previously used in the literature for studying neural networks (lines 140-144), in my understanding this model still relies on a Taylor approximation for the loss function of a neural network (for non-linear models). It is therefore not clear how the error introduced through this approximation would translate into the convergence analysis presented in the submission
- The authors use additional simplifying assumptions, such as the fact that each node can compute the true gradient of its submodel, which would be infeasible in the case of large datasets. Additionally, the results are only presented for Perm-1 sketches when the number of nodes matches the dimension of the model, which is again not a practical use-case. While the authors argue that their results can generalize beyond these limitations, a more general formulation is not provided in the submission.
- Other works have analyzed the convergence guarantees for IST, notably [28] has done so for a one hidden layer network with ReLU activations, which is a more general case than the one from this submission. It is not clear what are the additional insights presented here, compared to the previous work.


**Questions:**

- Can the authors please detail how their choice of the quadratic loss function ties to practical applications of model parallelism (e.g. for neural networks)?
- In Equation (12), do the matrices $L_i$ also need to be semi-positive definite?
- For Theorems 1 and 2, can the authors please describe how the results would change when using an unbiased gradient estimator, instead of the full gradient for each node?
- Are there any situations, without preconditioning, which would satisfy the conditions from Theorem 1, for model parallelization? The Identity example from Section 3.1 seems to only apply to data parallelization.
- For Equation (23), I am confused by the notation for the scaling coefficient. Should it be $[(L_i)_{\pi_i, \pi_i}]^{-1/2 }$ instead?
- From the analysis in Section 3.1.2 it looks like the fact that $C_i$ is biased does not have any effect on the bound. Can the authors provide more insights on why that is the case?
- In Appendix B.4. the authors show generalizations for $d>n$ and for different sketches. I think these would be useful to highlight in the main submission, since they correspond to a more practical setting.


**Limitations:**

I believe the authors have properly addressed the limitations of their analysis. However, I am not convinced that the contributions presented in this submission are broad enough, which ultimately motivates my score.


---------------------------------
**Edited after rebuttal**

After reading the authors' answers, and the other reviews, I decided to increase my overall rating to 5, as well as the scores for Soundness, Presentation and Contribution.

---

> ### Author Rebuttal · Authors · 2023-08-07
>
> Dear reviewer 87cd,
>
> Thanks for your time and effort. Let us respond to mentioned weaknesses and questions separately.
> # Responses to questions
> > Can the authors please detail how their choice of the quadratic loss function ties to practical applications of model parallelism (e.g. for neural networks)?
>
> The quadratic loss function is one of the most common (along with cross-entropy) choices of loss functions in supervised machine learning. Neural networks are not an exception.
> Please note that some of the reasons why the quadratic model was chosen are listed in the main paper.
> We would like to bring your attention to the fact that even for such a “simple” problem, the analysis is very non-trivial because the resulting gradient estimator is biased. We show that even for an interpolation homogeneous case, the algorithm may not converge. Our main goal is to study the properties of the method used for training large models with combined model and data parallelism. We believe that the quadratic problem is very well demonstrative for our purposes. In addition, as of today, we are not aware of any optimization theory for deep neural networks which does not simplify the actual practical settings.
>
> > In Equation (12), do the matrices $L_i$ also need to be semi-positive definite?
>
> No, it is not necessary. For the Th. 1 we require the average matrix $\bar{\mathbf{L}} = \frac{1}{n} \sum_{i=1}^n \mathbf{L}_i$ to be positive definite. Later for heterogeneous sketch preconditioning (Sec. 3.1.2), we also need the existence of $\mathbf{D}_i^{-1/2}$ for $\mathbf{D}_i = \mathrm{Diag}(\mathbf{L}_i)$, which is more general than semi-positive definiteness of every $\mathbf{L_i}$.
>
> > For Theorems 1 and 2, can the authors please describe how the results would change when using an unbiased gradient estimator, instead of the full gradient for each node?
>
> We would like to note that the current question form probably does not allow a proper answer, as we are not aware of any analysis of Stochastic Gradient Descent (SGD)-type methods with only an unbiasedness assumption. Although, if the gradient estimator $g(x)$ enjoys a bounded variance property (one of the most used in the stochastic optimization literature) $\mathbb{E} \||g(x) - \mathbb{E} g(x) \||^2 \leq \delta^2$ then our results can be easily extended. Namely, such local gradient estimators will introduce an additional neighborhood term $\gamma \delta^2$ in our convergence bounds. This can be obtained using the bias-variance decomposition equation (38).
>
> > Are there any situations, without preconditioning, which would satisfy the conditions from Theorem 1, for model parallelization?
>
> No, we are not aware of any such cases beyond the ones studied in our work. We want to note that if random (instead of permutation) sparsification is used, it will lead to similar challenges related to satisfying inequality (14) and Remark 1.
>
> > For Equation (23) ... should it be $[(L_i)_{\pi_i, \pi_i}]^{-1/2 }$ instead?
>
> Yes, you are correct. Thank you for spotting this. We have already fixed it in the revision of the paper.
>
> > Can the authors provide more insights on why biased $C_i$ does not have any effect on the bound in Section 3.1.2?
>
> At first, we found this result quite surprising as such good convergence bounds indeed seem unusual for a method with a biased gradient estimator. To share some insight and our intuition, we would like to note that our sketch preconditioning results in the following gradient estimator
> $\mathbb{E} g^k = \mathbb{E} \overline{\mathbf{B}}^k x^k = x^k,$
> which can be viewed as an optimal preconditioning for the true gradient of the quadratic (interpolated) problem $\overline{\mathbf{L}}^{-1} \nabla f(x^k) = \overline{\mathbf{L}}^{-1} \overline{\mathbf{L}} x^k = x^k$. Thus, the standard permutation sketch only leaves (as non-zeros) the diagonal elements, which are then scaled by our modification with local smoothness matrices $\mathbf{L}_i$.
>
> > I think these would be useful to highlight generalizations for $d>n$ and for different sketches in the main submission, since they correspond to a more practical setting.
>
> Thank you for the suggestion. We originally left this material in the Appendix in order to simplify the presentation of the key results.
>
> ## Comments on Weaknesses
> > additional simplifying assumptions, such as the fact that each node can compute the true gradient of its submodel
>
> Please note that we made these simplifications with a particular purpose in mind. Our goal was not to try to analyze the closest to a practical setting problem but rather to focus on particular new (and challenging) properties of the considered formulation. Quite often, such “closest to a practical” approaches, unfortunately, lead to very loose (sometimes even vacuous) bounds, which are obtained under restrictive (hard to check) assumptions introduced to facilitate the analysis, such as, i.e., bounded gradient norm and strong convexity (almost conflicting conditions).
>
> We believe that the contribution of this work includes a formalization of a novel theoretical setting, which basically has not been studied before. That is why it was essential for us to “isolate” the effects of homogeneous/heterogeneous distribution and computations with respect to submodels. To illustrate our position, let us refer to breakthrough optimization works on clipping [1] and local methods [2] (in addition to those mentioned in our paper works on GD with delayed updates and cyclical step-sizes), which considered simple (full) gradient descent updates which allowed to focus on the particular challenges of the considered problem formulations and provided insights which lead to improved understanding of the corresponding methods.
> ___
> [1] Zhang, Jingzhao, et al. "Why Gradient Clipping Accelerates Training: A Theoretical Justification for Adaptivity." ICLR 2019.
>
> [2] Khaled, Ahmed, et al. "First analysis of local GD on heterogeneous data." arXiv preprint:1909.04715 (2019).
>
> Regards, Authors

---

> > ### Comment · Reviewer_87cd · 2023-08-15
> >
> > Thank you for your answers and clarifications! I have increased my rating from 4 to 5. Although I agree that this is a thorough theoretical analysis, I still believe the overall setting is limited. I would like to ask the authors to include in the next revision a more comprehensive discussion on how the insights from the current theoretical analysis could be used in a more practical and general setting.

---

### Author Rebuttal · Authors · 2023-08-07

Dear Reviewers,

Firstly, we would like to thank the reviewers for their time reading our paper and for their feedback! This is all much appreciated.

We are encouraged that the considered problem is found _important and practical_, needing theoretical insights (R. 87cd). We are pleased that the limitations we highlighted are deemed _useful to be aware of_ (R. 87cd). We are glad our analysis is recognized as a _solid analytical treatment_ (R. LM8J). Reviewers ijy7 and LM8J also acknowledged the _clarity of writing and good presentation_ of the material.

Concerning the comparison with the work of **Liao and Kyrillidis [28]**.

We believe our works are so different that it does not make sense to include a detailed description of distinctions in our submission. After reading the reviews, we may incorporate the below response into the Appendix of the revised version of our work. Next, we would like to give detailed comments on the differences.

`Disclaimer:` we try our best to briefly and accurately represent some of the previous work’s findings.

### Assumptions
- The authors of [28] consider a “Single Hidden-Layer Neural Network with ReLU activations” and that the network’s first layer weights are initialized based on $\mathcal{N} (0, \kappa^2 \mathbf{I})$ and weight vector of the second layer is initialized uniformly at random from $\{−1, 1\}$. In contrast, we do not make any assumptions on the initialized parameters $x$ (in our notation).

- The second differentiation is assumptions on the data. The paper [28] assumes that for every data point $(a_j, y_j)$, it holds that $\||a_j\||^2 = 1$ and $|y_j| \leq C-1$ for some constant $C \geq 1$. Moreover, for any $j \neq l$, it holds that the points $a_i, a_l$ are not co-aligned, i.e., $a_i  \neq \xi a_l$ for any $\xi \in \mathbb{R}$. In contrast, we do not make any assumptions about the data apart from the ones on matrices $\mathbf{L}_i$.

- In addition, analysis in [28] assumes that the number of hidden nodes is greater than a certain quantity and that NN’s weights distance from initialization is uniformly bounded.

### Model

The authors of [28] consider a **regression (MSE) loss** function, a special case of quadratic loss and **full gradients** computation. They provide guarantees for IST under a “s​​implified assumption that every worker has full data access”, which corresponds to the homogeneous setting in our terminology.

### Analysis

Their analysis is based on the Neural Tangent Kernel (NTK) framework, which typically relies on first-order Taylor approximation of the neural network output. This approximation depends on finite-width NTK matrix $H(k)$ at iteration $k$. In the overparameterized regime, the change of $H(k)$ is small, staying close to the NTK at initialization, approximated by the infinite-width NTK. In the end, such a series of relaxations leads to the following **quadratic loss approximation**:
$$
f_{k+1} \approx f_k + \left\langle\nabla_{\mathbf{u_k}} f_k, \mathbf{u_{k+1}}-\mathbf{u_k}\right\rangle
\approx
f_k - \xi \eta\left\langle\mathbf{u_k} - \mathbf{y}, \mathbf{H}(k)\left(\mathbf{u_k} - \mathbf{y}\right)\right\rangle,
$$
which combined with the assumption on the minimum eigenvalue $\lambda_{\min }(\mathbf{H}(k)) \geq \frac{\lambda_0}{2} > 0$ leads to a linear convergence result for a strongly-convex problem.

Besides, results in [28] show linear convergence of the model weights to the neighborhood of the solution, which is typical for strongly convex and smooth problems. Wherein our results focus on reaching the stationary point (although we also provide some bounds for the iterates and loss convergence), which is more common in the non-convex settings. Another difference is that they provide high probability convergence analysis.

We hope that there will be a productive discussion. Please do not hesitate to ask any further questions!

Best regards,

Paper 6457 Authors

___

[28] Fangshuo Liao and Anastasios Kyrillidis. On the convergence of shallow neural network training with randomly masked neurons. Transactions on Machine Learning Research, 2022.

---

### Decision · Program_Chairs · 2023-09-21

**Decision:**

Reject

**Comment:**

The submission presented a new theoretical analysis for algorithms performing independent subnetwork training (IST), under various distribution setups, providing convergence guarantees in the quadratic case.

The reviewers agreed that the submission tackles a hard theoretical problem and that the results appear sound, but were less convinced by the narrow nature of the analysis, and lack of practical implications. The paper was therefore borderline; unfortunately, the decision tended negative given the highly-competitive nature of NeurIPS this year.

The AC would encourage the authors to broaden the paper's contribution (potentially shoring up the practical validation), and re-submit to a future venue.